# R&D-Agent: An LLM-Agent Framework Towards Autonomous Data Science

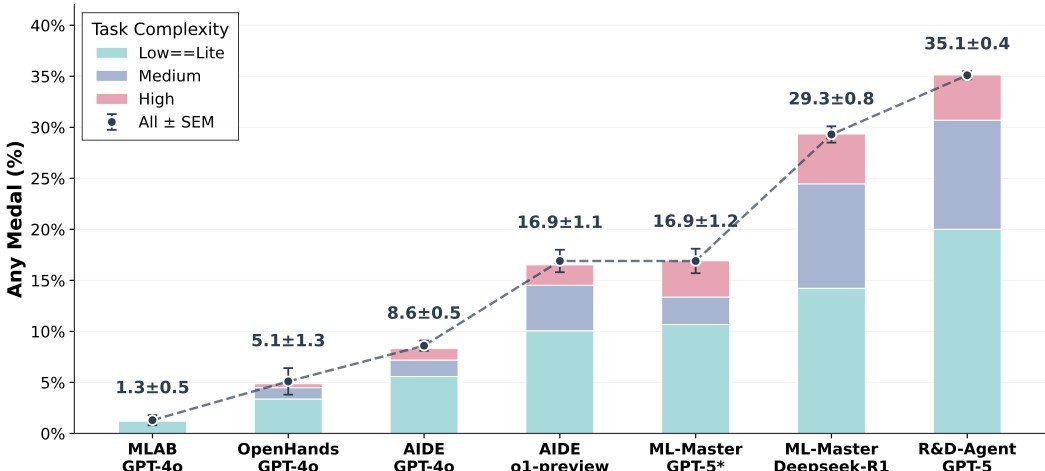

Figure 1: Agent performance on MLE-Bench. Stacked bars show any medal rates for Low==Lite (22 tasks), Medium (38 tasks), and High (15 tasks) complexity levels. The dashed line indicates overall performance (mean ± SEM). R&D-Agent achieves SOTA performance at 35.1± 0.4%. * indicates our re-evaluation of ML-Master within our environment.

## Abstract

Recent advances in AI and ML have transformed data science, yet increasing complexity and expertise requirements continue to hinder progress. Although crowd-sourcing platforms alleviate some challenges, high-level machine learning engineering (MLE) tasks remain labor-intensive and iterative. We introduce R&D-Agent, a comprehensive, decoupled, and extensible framework that formalizes the MLE process. R&D-Agent defines the MLE workflow into two phases and six components, turning agent design for MLE from ad-hoc craftsmanship into a principled, testable process. Although several existing agents report promising gains on their chosen components, they can mostly be summarized as a partial optimization from our framework's simple baseline. Inspired by human experts, we designed efficient and effective agents within this framework that achieve state-of-the-art performance. Evaluated on MLE-Bench, the agent built on R&D-Agent ranks as the top-performing machine learning engineering agent, achieving 35.1% any medal rate, demonstrating the ability of the framework to speed up innovation and improve accuracy across a wide range of data science applications.

## 1 Introduction

Over the past decade, artificial intelligence (AI) and machine learning (ML) have fundamentally reshaped data science, driving advances across domains as diverse as machine translation (Isik et al., 2025), recommendation systems (Yuan et al., 2025), social simulation (Yang et al., 2025), and medical diagnostics (Sepehri et al., 2025). The growing availability of large-scale datasets (Ghorbani et al., 2023), coupled with rapid algorithmic progress, has enabled models that deliver increasingly accurate and adaptive outcomes. However, as data becomes more heterogeneous and high-dimensional, so

does the demand for experienced data scientists who can craft appropriate models, interpret nuanced patterns, and iterate toward optimal solutions.

Crowdsourcing platforms like Kaggle[1] partially mitigate the expertise bottleneck by mobilizing thousands of data scientists, yet they also expose the limits of human-driven workflows: even top teams spend considerable time on trial-and-error experimentation, feature crafting, and hyperparameter tuning, making progress labor-intensive and slow.

Large language models (LLMs) offer an opportunity to mitigate these limitations. Their demonstrated strengths in code generation and reasoning(Achiam et al., 2023; Team et al., 2023; Liu et al., 2024) suggest they can automate exploration of large design spaces. Machine Learning Engineering (MLE) serves as an ideal testbed for this potential, as success critically depends on efficiently navigating vast configuration spaces to identify optimal solutions. However, a significant gap persists between promise and practice. Benchmarks like MLE-bench (Chan et al., 2024), built on real Kaggle competitions, report that state-of-the-art LLMs reach only a small fraction of human expert performance, whereas well-designed agents can perform much better, highlighting the potential of improved agent design to accelerate MLE exploration.

Inspired by the common workflow of data scientist, we introduce R&D-Agent, a comprehensive, decoupled, and extensible framework that formalizes the MLE process. Mirroring how data scientists work in practice, R&D-Agent separates: (i) a **Research phase** focused on idea generation and search-covering *planning* (e.g., dynamically adjusting guidelines over time), *exploration path structures* (e.g., tree-based search vs. chain-based search), *memory context* (organizing and retrieving prior solutions and knowledge), and *reasoning pipelines* (for hypothesis formation and refinement); and (ii) a **Development phase** focused on implementation and feedback-covering *coding workflows* (steps turning ideas into runnable code efficiently) and *evaluation strategy* (obtaining reliable and robust data-driven feedback). For each aspect, a simple and LLM-first baseline strategy is established as a clear starting point.

Positioning existing systems within this decomposition clarifies their coverage and gaps. As shown in Table 1, previous work typically optimizes only a narrow slice of the workflow. For instance, AIDE (Jiang et al., 2025) and ML-Master (Liu et al., 2025) primarily target the *exploration path structure* via tree-based search, while MLE-STAR (Nam et al., 2025) explores deeply along a single chain. KompeteAI (Kulibaba et al., 2025) emphasizes the *coding workflow* with faster debugging. **All the existing methods can be summarized as a partial optimization from our framework's simple baseline.**

Table 1: Comparison of R&D-Agent with existing methods, grouped by **Research** and **Development** phase design aspects. "/" denotes not covered or a simple, LLM-first baseline strategy. Research phase includes: (i) *Planning*, (ii) *Exploration Path Structuring*, (iii) *Memory Context*, (iv) *Reasoning Pipeline*. Development phase includes: (v) *Coding Workflow*, (vi) *Evaluation Strategy*. ✿ indicates that the framework provides multiple options followed by recommended best practices. Details of R&D-Agent's design are provided in Sec. 3.3.

| Phase | Design Aspect | Agent Designs | | | | Frameworks | |
|---|---|---|---|---|---|---|---|
| | | AIDE (Jiang et al., 2025) | ML-Master (Liu et al., 2025) | KompeteAI (Kulibaba et al., 2025) | MLE-STAR (Nam et al., 2025) | AIRA (Toledo et al., 2025) | **R&D-Agent (Ours)** |
| **Research** | *Planning* | / | / | / | / | / | ✿ Dynamic |
| | *Path Structuring* | Tree (Greedy) | Tree + MCTS | Tree + Merging | Chain | ✿ Tree + MCTS | ✿ Adaptive |
| | *Memory Context* | / | Sibling | Sibling | Sibling | ✿ Sibling | ✿ Collab. Comm. |
| | *Reasoning Pipeline* | One step | One step | One step | One step | One step | ✿ Scientific multi-step |
| **Development** | *Coding Workflow* | Node Debug | Node Debug | Debug | Debug | / | ✿ Eff. & Iter. Debug |
| | *Evaluation Strategy* | / | / | / | / | / | ✿ Aggregated |

Although these systems report promising gains on their chosen components, they often (i) cover only a subset of the full MLE workflow, (ii) entangle multiple steps into monolithic pipelines rather than cleanly separating them into specialized components or agents, and (iii) leave many alternative designs within each phase unexplored. As a result, insights and improvements are difficult to generalize or reuse across tasks and domains. While Toledo et al. (2025) recognize the need for a flexible framework, their design offers limited agent-level modularity and restricts exploration of the broader design space.

---

[1]https://www.kaggle.com/

Our framework directly addresses these issues. R&D-Agent (i) is comprehensive, enabling systematic exploration across the entire MLE workflow; (ii) is well-decoupled, allowing each phase to be implemented, replaced, and improved by dedicated components or expert agents; and (iii) is highly extensible, supporting plug-and-play alternatives that unify and generalize prior systems while facilitating the discovery of new agent configurations. In short, R&D-Agent turns agent design for MLE from ad-hoc craftsmanship into a principled, testable process.

Guided by human expertise, we instantiate R&D-Agent and conduct systematic ablations across each phases to isolate the contribution of each component. Composing the best-performing choices shown in Table 1 within the framework yields a well-configured agent that delivers significant gains on MLE-bench (Chan et al., 2024), achieving new state-of-the-art results.

In summary, our primary contributions are:

- We introduce R&D-Agent, a comprehensive, decoupled, and extensible autonomous agent framework that formalizes the MLE process by separating a research phase (planning, exploration path structure, memory context, and reasoning pipelines) from a development phase (coding workflows and evaluation strategy).

- With R&D-Agent enabling plug-and-play alternatives, we conduct systematic ablations across both phases to isolate the contribution of each component and derive insights into the factors that most affect agent performance on data science.

- Guided by human expertise, an R&D-Agent configuration that composes the best-performing choices within the framework delivers significant gains on MLE-bench (Chan et al., 2024), achieving new state-of-the-art results and demonstrating the efficiency and effectiveness of our approach.

## 2 PREREQUISITES

In the context of MLE agents, the main goal is to maximize the evaluation score on a given benchmark under a limited time budget. Formally, let $\mathcal{T}$ be an ML task with dataset $\mathcal{D} = \{\mathcal{D}_{\text{dev}}, \mathcal{D}_{\text{test}}\}$, where $\mathcal{D}_{\text{dev}}$ is the development set and $\mathcal{D}_{\text{test}}$ is the test set. The agent can only develop, tune, and iterate its solution on $\mathcal{D}_{\text{dev}}$, while the final performance is measured on the unseen $\mathcal{D}_{\text{test}}$. This fundamental separation means the agent never has access to the final evaluation metric during its development process. Instead, it must rely on a proxy metric, $M(s; \mathcal{D}_{\text{dev}})$, evaluated on the development set to guide its exploration. Let $T$ denote the total allowed wall-clock time for the agent to generate a complete solution (including the actual execution/running time of the solution code). Let $s = \text{Agent}(\mathcal{T}, T)$ be a candidate solution generated by the agent (e.g., a Python script for data preprocessing, model training, and evaluation) within this generation-time budget $T$. The performance of $s$ on $\mathcal{T}$ is measured by a task-specific metric $M(s; \mathcal{D}_{\text{test}}) \in \mathbb{R}$, such as accuracy or correlation.

The agents objective can be written as the constrained optimization problem

$$s^{\star} = \arg \max_{\substack{s = \text{Agent}(\mathcal{T}, T), \\ s \in \mathcal{S}}} M(s; \mathcal{D}_{\text{test}}), \quad \text{s.t.} \quad \text{gen\_time}(s) \leq T,$$

where $\mathcal{S}$ is the set of all feasible solutions the agent can generate and gen_time($s$) denotes the total generation time spent by the agent to produce $s^{\star}$.

## 3 R&D-AGENT

The architecture of the R&D-Agent framework, illustrated in Figure 2, addresses the core challenge of efficiently exploring optimal solutions in MLE. Built on a modular design principle, it decomposes the entire pipeline into distinct, configurable components. The framework orchestrates two specialized agents: a Research Agent that explores ideas through parallel paths, and a Development Agent that implements and iteratively refines these proposals.

This modularity transforms agent development from monolithic construction into compositional optimization, enabling systematic exploration of a vast configuration space. Guided by human expert workflows, we navigated this space to identify an optimal design that achieves state-of-the-art performance on MLE-Bench, validating both our modular approach and the effectiveness of human-inspired reasoning patterns.

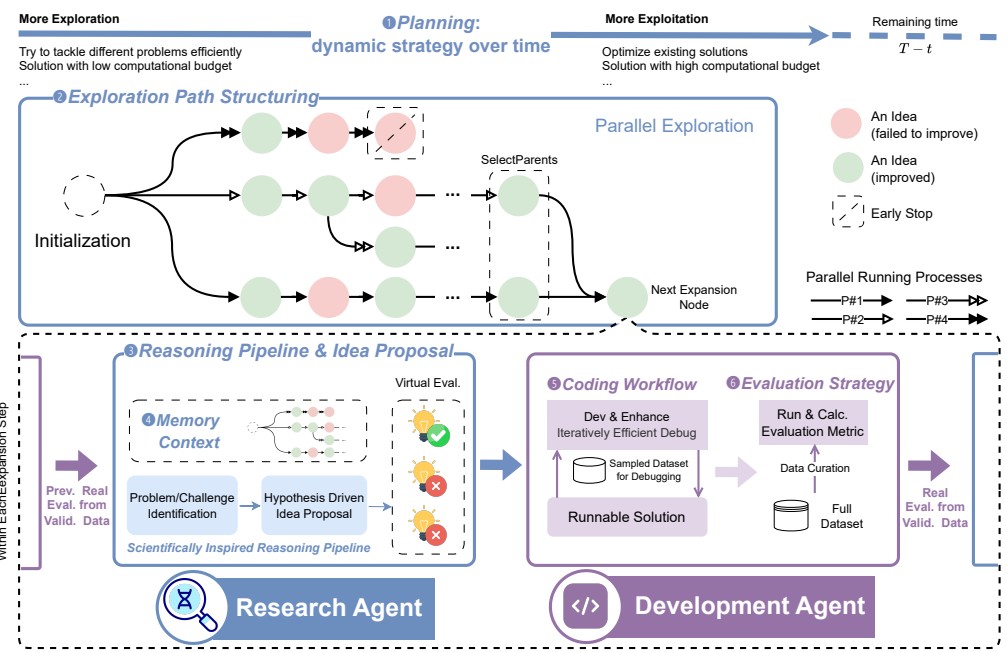

Figure 2: Framework of R&D-Agent. R&D-Agent works in an iterative loop in which the Research Agent proposes ideas and the Development Agent implements them into runnable solutions to obtain feedback from data. By decoupling high-level research from low-level implementation, the framework efficiently explores the solution space through parallel exploration paths and iterative refinement, progressively converging on optimal solutions.

## 3.1 FRAMEWORK OVERVIEW

Inspired by the common workflow of data scientists, the R&D-Agent framework defines six key, extensible components, organized into two main phases: the Research Phase and the Development Phase. We introduce the Framework Concept (FC) of each component in this section.

**The Research Phase.** The research phase aims to discover and refine promising ideas before committing to costly implementation. It consists of four primary components:

❶ **FC-Planning:** This component focuses on the high-level allocation of exploration effort over time. Since data science tasks are sequential decision problems involving iterative trial-and-error, an effective plan must dynamically adjust the timing, budget, and guidelines for idea exploration. It adapts priorities and resources as new information emerges, managing the crucial trade-off between exploration and exploitation.

❷ **FC-Exploration Path Structuring:** This component determines how the solutions are organized and the historical solutions to which which they will be referred. Strategies range from greedy chain-based approaches (Nam et al., 2025), which offer fast convergence at the risk of local optima, to tree-based search methods (Liu et al., 2025; Kulibaba et al., 2025), which maintain greater diversity at a higher computational cost. Our framework supports hybrid and adaptive designs that can combine these strengths.

❸ **FC-Reasoning Pipeline:** This component defines how knowledge from the *memory context* is transformed into concrete research ideas. This process may include dataset analysis, hypothesis formulation, benefit justification, trade-off assessment, and implementable solution sketches. A clear, structured reasoning pipeline improves the quality, novelty, and feasibility of ideas, while supporting systematic evaluation before moving to development.

❹ **FC-Memory Context:** This component manages how accumulated knowledge, such as historical solutions, evaluation results, and insights, is stored, retrieved, and reused to inform the *reasoning pipeline*. Well-structured memory enables knowledge transfer between iterations, reduces redundant exploration, and stabilizes long-horizon reasoning.

**The Development Phase.** The development phase turns the most promising research ideas into fully bug-free and evaluated solutions. It is composed of two key components:

❺ **FC-Coding Workflow:** This component covers the process from a conceptual idea to bug-free code. It emphasizes modular design, efficient iterative debugging, and early detection of runtime issues. Techniques like rapid prototyping on sampled data can significantly shorten the development cycle while preserving correctness.

❻ **FC-Evaluation Strategy:** This component ensures reliable and consistent assessment of solution performance. A strong evaluation strategy involves choosing stable metrics, adhering to fixed validation settings, and using aggregated evaluations to reduce noise. This mitigates overfitting risks, filters out spurious high scores from underfitting, and ensures that iterative improvements reflect genuine performance gains rather than evaluation artifacts.

## 3.2 FRAMEWORK FORMULATION

We formalize the R&D-Agent framework with the high-level algorithm presented in Algorithm 1. The algorithm iteratively builds an exploration graph $\mathcal{G}$, by executing a loop composed of a Research Phase and a Development Phase until the time budget $T$ is met. Each component in the algorithm corresponds to the design aspects detailed in our framework overview. We call the loop in Algorithm 1 containing a research and development phase as **R&D loop**.

---

**Algorithm 1:** High-Level Algorithm of the R&D-Agent Framework

---

**Notation :** $\mathcal{G}_t$: exploration graph; $\pi_t$: plan; $\mathcal{N}_t$: parent nodes; $c_t$: context; $i_t$: idea; $x_t$: code; $s_t$: score.

**Input** : ML task $\mathcal{T}$, total time budget $T$

**Output** : Final solution $x^*$

---

$\mathcal{G} \leftarrow \emptyset$                           `// Initialize exploration graph`

**while** elapsed_time() $< T$ **do**

    `// Research Phase`

    $\pi \leftarrow \mathcal{P}(\mathcal{G}, \text{elapsed\_time}(), T)$                `// Planning`

    $\mathcal{N} \leftarrow \text{SelectParents}(\mathcal{G}, \pi)$       `// Exploration Path Structuring`

    $c \leftarrow \mathcal{M}(\mathcal{G}, \pi)$                      `// Memory Context`

    $i \leftarrow \mathcal{R}(c, \mathcal{N}, \pi)$                  `// Reasoning Pipeline`

    `// Development Phase`

    $x \leftarrow \text{Dev}(i, \mathcal{N})$                 `// Coding Workflow`

    $s \leftarrow \text{Eval}(x, \mathcal{T})$             `// Evaluation Strategy`

    $\mathcal{G} \leftarrow \mathcal{G} \cup \{(\mathcal{N}, i, x, s)\}$       `// Update exploration graph`

**end**

$x^* \leftarrow \text{Submit}(\mathcal{G})$         `// Select and submit final solution`

---

## 3.3 A HUMAN-EXPERT-INSPIRED AGENT DESIGN

The R&D-Agent framework enables the systematic discovery of novel agent architectures. Guided by the workflows of human experts, we identified a specific, highly efficient agent configuration that establishes a new state-of-the-art on MLE-Bench. This section details the design choices for each of the six components in this SOTA configuration, with its core ideas illustrated in Figure 2. We introduce the Module Design (MD) of each component in this section.

❶ **MD-Planning.** Like human experts in scientific exploration, we quickly identify promising directions in the early stages and move to more sophisticated solutions later. To achieve this, we use a dynamic planning strategy that adjusts over time. In the early stage (e.g., the first hour), the agent is given a limited computational budget, which discourages the use of heavy techniques such as ensembles or cross-validation. As promising directions emerge, the budget gradually increases, enabling more costly yet effective methods (e.g., ensembles, cross-validation) in later stages (e.g., at 4h). The plan also steers idea generation: early stages encourage novelty, while later stages focus on refining high-performing solutions with proven techniques.

❷ **MD-Exploration Path Structuring.** Like human experts, R&D-Agent explores multiple research directions in parallel and merges their strengths at the last stage for optimal solutions. We

adopt an adaptive DAG-based exploration structure guided by a core insight: initial implementations have disproportionate impact on exploration diversity, as subsequent steps are inherently path-dependent. Therefore, we maximize diversity in the first layer to establish distinct research directions, then greedily exploit the best solution within each branch while pruning sub-optimal paths. This design achieves efficient parallel exploration and result fusion.

❸ **MD-Scientific Reasoning Pipeline.** Human data scientists typically follow a rigorous reasoning process to propose research ideas, which is a valuable distillation of human intelligence. Inspired by this, we propose a scientific multi-step reasoning pipeline that begins by analyzing the current solution and dataset characteristics to identify the most critical problem (e.g., for a time-series dataset, mining temporal patterns is crucial for high performance). Rather than generating shallow ideas, the agent dives deeper into the problem, formulating hypotheses on why a proposed method would address it (e.g., an RNN can capture temporal dependencies in time-series data), and finally outputs an idea implementable by an LLM (e.g., training an LSTMGraves (2012) to model dependencies). Since verifying ideas is more costly than generating them, we introduce a virtual evaluation strategy: the agent generates multiple ideas during reasoning, uses LLM-based assessments to select the most promising one, and sends only that to the development phase.

❹ **MD-Memory Context.** We enhance collaborative memory to enable knowledge sharing across parallel branches without sacrificing diversity. Each branch begins with a distinct idea and explores independently. After hypothesis generation, we augment each branch's context with two sources: the best ideas across all branches and a probabilistically sampled subset from others. The sampling kernel favors ideas that are topically similar and have higher scores, with more recent ideas weighted higher. An LLM then selects the most promising candidates from this enriched pool, accelerating convergence while maintaining exploration diversity.

❺ **MD-Coding Workflow.** To improve efficiency, particularly for solutions with long runtimes, we adopt an efficient and iterative debug workflow. Our LLM-powered agent first samples a small, representative subset of the training data and enters a rapid prototyping loop: implementing the proposed idea, testing on this subset, and refining based on immediate feedback. This cycle continues until achieving a runnable and logically sound solution on the subset. Only validated solutions proceed to full-scale evaluation. This approach mirrors human rapid prototyping practices, significantly reducing development time by catching errors early and ensuring that computational resources are not wasted on flawed implementations. The workflow enables the agent to explore substantially more solution candidates within the same time budget.

❻ **MD-Evaluation Strategy.** To ensure robust and reliable performance assessment, we implement an aggregated evaluation strategy with standardized protocols. A key challenge in many MLE tasks is that evaluation logic is part of the agent's solution, causing inconsistent metrics and data splits that prevent fair comparison. We address this through two mechanisms. First, we enforce standardized data splitting: preparing fixed train-validation-test splits at the beginning of agent runs, with test data remaining entirely inaccessible for final grading. Second, beyond standard validation-based selection, we introduce an additional evaluation layer that collects top solutions from different exploration branches and evaluates them using consistent metrics on the same validation set. This aggregated approach ensures fair comparison across diverse solution strategies and enables more reliable selection of the best-performing solution for final submission.

## 4 EXPERIMENT

To rigorously evaluate R&D-Agent's capabilities in realistic data science scenarios, we conduct extensive experiments on MLE-Bench (Chan et al., 2024). As a benchmark comprising a diverse set of authentic Kaggle competitions, MLE-Bench serves as an excellent proxy for real-world challenges, demanding a holistic combination of strategic thinking and robust engineering for agents to autonomously design, build, and train models.

### 4.1 EXPERIMENT SETUP

All our experiments are conducted on the MLE-Bench (Chan et al., 2024) benchmark. We compare R&D-Agent against leading open-source systems, primarily ML-Master (Liu et al., 2025) and AIDE (Jiang et al., 2025), using their official leaderboard metrics. In time budget, we align with ML-Master allowing 12 hours compared to the official 24 hours setting. In computation, our environment

has a lower throughput with 12 vCPUs, 220GB RAM, and 1 V100 GPU while AIDE uses 36 vCPU, 440GB RAM and 1 A10 GPU, ML-Master uses 36 vCPU, 512GB shared RAM and 1 A100 GPU. Therefore, our experimental setup is intentionally more **challenging** than all the previous work.

To ensure fair comparison, we evaluate R&D-Agent using two frontier LLM configurations: (1) GPT-5 only, and (2) a hybrid o3(R) + GPT-4.1(D), where o3 powers the Research phase and GPT-4.1 the Development phase. Furthermore, We re-evaluated the previous SOTA, ML-Master, under our identical 12-hour setting using both configurations. All our new results are averaged over three runs with different random seeds for statistical robustness. Our evaluation is based on the official MLE-Bench metrics, with the Any-Medal Rate as the primary indicator of overall performance.

## 4.2 MAIN RESULTS

R&D-Agent establishes a new SOTA on MLE-Bench (Chan et al., 2024), significantly outperforming all existing open-source systems. As illustrated in Figure 1, R&D-Agent powered by GPT-5 achieves 35.1% Any-Medal Rate, exceeding the previous state-of-the-art ML-Master (Liu et al., 2025) (29.3% with Deepseek-R1) by 5.8 percentage points. The stacked bars further reveal R&D-Agent's consistent superiority across all task complexity levels, demonstrating its robust performance on diverse data science challenges.

Table 2 further demonstrates our framework's architectural advantages. With GPT-5, R&D-Agent achieves the highest Any-Medal Rate at $35.1 \pm 0.4\%$, while our hybrid o3(R) + GPT-4.1(D) configuration also excels at $29.7 \pm 0.4\%$, with both substantially outperforming all prior systems. Crucially, when ML-Master was re-evaluated with GPT-5 in our environment, it achieved only $16.9 \pm 2.0\%$[2]. This direct comparison under identical LLM configurations confirms that our framework's design, not merely model selection, drives the substantial performance gap. Both R&D-Agent configurations demonstrate strong performance across multiple metrics, with GPT-5 achieving $45.3 \pm 0.0\%$ Above-Median Rate and $16.4 \pm 0.9\%$ Gold Medal Rate, showcasing the framework's ability to consistently produce competitive solutions regardless of the underlying LLM choice.

Table 2: Comparative performance of all agents across the official MLE-Bench evaluation metrics. All results represent the mean $\pm$ SEM from three independent runs with different random seeds. The top-performing agent is highlighted in **bold** and the second-best is underlined. * indicates our re-evaluation of ML-Master (Liu et al., 2025) within our environment (V100 GPU) to ensure fair comparison under identical conditions. Complete individual run results are provided in Appendix D.1.

| Agent | Valid Submission (%) | Above Median (%) | Bronze (%) | Silver (%) | Gold (%) | Any Medal (%) |
|---|---|---|---|---|---|---|
| **MLAB (Huang et al., 2023)** | | | | | | |
| GPT-4o | $44.3 \pm 2.6$ | $1.9 \pm 0.7$ | $0.0 \pm 0.0$ | $0.0 \pm 0.0$ | $0.8 \pm 0.5$ | $0.8 \pm 0.5$ |
| **OpenHands (Wang et al., 2024)** | | | | | | |
| GPT-4o | $52.0 \pm 3.3$ | $7.1 \pm 1.7$ | $0.4 \pm 0.4$ | $1.3 \pm 0.8$ | $2.7 \pm 1.1$ | $4.4 \pm 1.4$ |
| **AIDE (Jiang et al., 2025)** | | | | | | |
| GPT-4o | $54.9 \pm 1.0$ | $14.4 \pm 0.7$ | $1.6 \pm 0.2$ | $2.2 \pm 0.3$ | $5.0 \pm 0.4$ | $8.7 \pm 0.5$ |
| o1-preview | $82.8 \pm 1.1$ | $29.4 \pm 1.3$ | $3.4 \pm 0.5$ | $4.1 \pm 0.6$ | $9.4 \pm 0.8$ | $16.9 \pm 1.1$ |
| **ML-Master (Liu et al., 2025)** | | | | | | |
| Deepseek-R1 | $93.3 \pm 1.3$ | $\underline{44.9 \pm 1.2}$ | $4.4 \pm 0.9$ | $\underline{7.6 \pm 0.4}$ | $\mathbf{17.3 \pm 0.8}$ | $29.3 \pm 0.8$ |
| o3(R) + GPT-4.1(D)* | $\mathbf{98.2 \pm 0.9}$ | $25.8 \pm 1.9$ | $5.8 \pm 1.6$ | $3.1 \pm 1.9$ | $9.3 \pm 0.8$ | $18.2 \pm 1.9$ |
| GPT-5* | $85.3 \pm 3.5$ | $26.2 \pm 1.6$ | $4.4 \pm 1.2$ | $3.1 \pm 0.4$ | $9.3 \pm 0.8$ | $16.9 \pm 1.2$ |
| **R&D-Agent (Ours)** | | | | | | |
| o3(R) + GPT-4.1(D) | $94.2 \pm 0.4$ | $\underline{44.9 \pm 0.4}$ | $6.2 \pm 0.9$ | $7.5 \pm 1.2$ | $16.0 \pm 0.8$ | $\underline{29.7 \pm 0.4}$ |
| GPT-5 | $\underline{96.0 \pm 0.0}$ | $\mathbf{45.3 \pm 0.0}$ | $\mathbf{6.7 \pm 1.5}$ | $\mathbf{12.0 \pm 0.8}$ | $\underline{16.4 \pm 0.9}$ | $\mathbf{35.1 \pm 0.4}$ |

---

[2]This differs from ML-Master's official 29.3% primarily due to: (1) hardware differences (V100 vs. A100 GPU), and (2) backend model differences (GPT-5 vs. Deepseek-R1). We report this for transparent comparison under identical conditions.

## 4.3 ABLATION STUDY

We quantify each component's contribution by removing one component at a time while preserving all others. For computational efficiency, evaluations are conducted on a curated subset of 40 competitions from MLE-Bench (Chan et al., 2024) using GPT-5 (see Appendix D.4). This subset includes tasks where our method, AIDE (Jiang et al., 2025), or ML-Master (Liu et al., 2025) achieved medals, ensuring coverage of scenarios most influenced by our design.

Our analysis aligns with the dual-phase architecture: the **research phase ablation** evaluates four components (*dynamic planning*, *exploration path structuring*, *memory context*, and *reasoning pipeline*), while the **development phase ablation** focuses on implementation elements (*coding workflows* and *evaluation strategy*).

### 4.3.1 RESEARCH PHASE ABLATION STUDY

Table 3 summarizes our analysis across four key metrics: (1) **Avg. Loops**: number of R&D loops per competition, (2) **Improve Rate**: percentage of R&D loops yielding performance gains, (3) **First-Medal**: time to first medal-winning solution, (4) **Medal Rate**: percentage achieving medals in the 40-competition subset. (5) **Any Medal**: percentage achieving medals in the whole 75-competition set. The 35 remaining competitions are all considered non-medal in this calculation.

Table 3: Research Phase Ablation Results on 40-competition subset. Each column removes one component while preserving others. Full System shows mean $\pm$ SEM over 3 runs; ablation results report single representative runs due to computational constraints.

| Metric | Full System | w/o Planning | w/o Exploration Path | w/o Reasoning Pipeline | w/o Memory Context |
|---|---|---|---|---|---|
| Avg. Loops | $45.9 \pm 2.2$ | 48.4 | 19.4 | 55.3 | 44.0 |
| Improve Rate (%) | $41.1 \pm 0.4$ | 39.4 | 40.0 | 23.0 | 40.9 |
| First-Medal (h) | $2.9 \pm 0.1$ | 1.7 | 3.0 | 1.7 | 2.0 |
| Medal Rate (%) | $65.8 \pm 0.8$ | 50.0 | 47.5 | 50.0 | 60.0 |
| Any Medal (%) | $35.1 \pm 0.4$ | 26.7 | 25.3 | 26.7 | 32 |

Each ablation reveals how R&D-Agent degrades toward existing baselines (Table 1), confirming our framework's architectural advantages:

- **w/o Planning (24% relative decline).** Degrading component *Planning* from *Dynamic* to baseline approaches lacking strategic resource allocation ("/"). Any Medal Rate drops from 35.1% to 26.7%. Despite faster First-Medal time (2.9h→1.7h), the system rushes to local optima without temporal adaptation, matching the performance degradation observed in methods like AIDE.

- **w/o Exploration Path (28% relative decline).** The most severe degradation occurs when degrading *Adaptive* path structuring to sequential chain exploration (similar to MLE-STAR (Nam et al., 2025)). Any Medal Rate drops to 25.3% and average loops fall dramatically (45.9→19.4), demonstrating how tree-based adaptive exploration fundamentally outperforms rigid chain structures in solution space coverage.

- **w/o Reasoning Pipeline (24% relative decline).** Degrading our *Scientific multi-step* reasoning to baseline *one-step* approaches used by prior methods. Any Medal Rate drops to 26.7% and Improve Rate collapses (41.1%→23.0%). Despite high exploration volume (55.3 loops), the system generates improvements in only 23% of loops, highlighting how structured multi-step reasoning enables more effective hypothesis generation than simple one-step approaches.

- **w/o Memory Context (9% relative decline).** Degrading component *Memory Context* from *Collaborative Communication* to a simple memory management approach(similar to ML-Master (Liu et al., 2025)), showing the smallest degradation to 32.0% Any Medal Rate. The preserved iteration efficiency suggests that while collaborative memory provides optimization benefits, core learning mechanisms remain functional through alternative pathways, validating our architectural separation between fundamental and enhancement components.

These results demonstrate that R&D-Agent's superior performance stems from architectural innovations, with exploration path structuring providing the most critical advantage over existing approaches.

### 4.3.2 DEVELOPMENT PHASE ABLATION STUDY

Unlike research components that affect idea generation, development components impact solution implementation quality and reliability. We analyze their contributions through temporal medal acquisition patterns over 12 hours, as shown in Figure 3. This temporal approach reveals not just final performance, but also *when* and *how quickly* each component contributes to success. Results combine ablation studies with one representative run from the primary experiment, illustrating the temporal dynamics of development components.

**Perfect Selection (Upper Bound).** Theoretical maximum performance if we could submit all generated solutions and select the best performer. This oracle bound reaches 37.3% Any Medal Rate, establishing our solution pool's upper limit.

**Full System.** Exhibits rapid initial progress (0-2h), steady refinement (2-6h), and convergence at 34.7% Any Medal Rate. Achieving 93% of perfect selection demonstrates effective solution identification without test set access.

**W/o Coding Workflow.** Degrading our *Efficient & Iterative Debug* workflow forces the system to use full-dataset debugging like baseline methods. Performance drops immediately to 24.0% Any Medal Rate and remains persistently degraded. This demonstrates that our sample-based debugging fundamentally outperforms traditional full-dataset approaches, where computational overhead severely constrains exploration within time budgetsthe same bottleneck that limits existing systems.

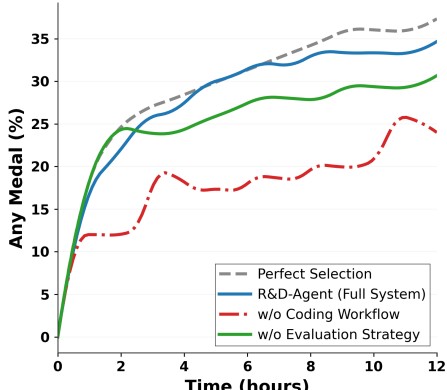

Figure 3: Development Phase Temporal Ablation. Medal acquisition rate (percentage of 75 competitions) over 12 hours reveals when and how each component contributes.

**W/o Evaluation Strategy.** Degrading our *Aggregated* evaluation strategy to baseline approaches without systematic evaluation. Performance initially matches the full system until hour 2, then diverges significantly to 30.7% Any Medal Rate. This delayed degradation shows that while basic evaluation suffices for simple solutions, sophisticated multi-dimensional evaluation becomes critical as solution complexity and over-fit risk increases-an advantage absent in existing methods.

These temporal patterns show how R&D-Agent's development innovations address fundamental bottlenecks overlooked by prior work: coding workflow provides implementation efficiency from the start, while evaluation strategy enables sustained improvement as solutions grow sophisticated.

## 5 RELATED WORK IN APPENDIX A

## 6 CONCLUSION

In this paper, we introduced R&D-Agent, a comprehensive and decoupled framework that transforms MLE agent design from monolithic construction into systematic exploration. We proposed explicit phase separation between research and development as the core design principle, implemented through six extensible and modular components that enable efficient exploration of complex solutions.

Guided by human-expert workflows, we discovered an optimal configuration that achieves 35.1% Any-Medal Rate on MLE-Bench, establishing a new state-of-the-art despite operating with more limited computational resources than prior work. Comprehensive ablation studies validate the contribution of each component, confirming that our framework's extensible architecture, rather than merely improved model capabilities, is the key driver of these performance gains.

R&D-Agent enables researchers to systematically test and compare different agent architectures within a unified framework, eliminating the need to rebuild entire systems from scratch. The framework's modularity allows precise attribution of performance gains to specific components, transforming agent development from trial-and-error into principled experimentation. By open-sourcing both the framework and our discovered configurations, we provide the ML community with immediately deployable solutions and a platform for further innovation in autonomous AI systems.

## REPRODUCIBILITY STATEMENT

To ensure reproducibility of our work, we provide comprehensive implementation details and resources throughout the paper and supplementary materials. The complete source code for the R&D-Agent framework, including all six modular components and our discovered optimal configuration, is available as anonymous supplementary material. Algorithm 1 presents the high-level framework structure, with detailed component implementations described in Section 3.3. Complete prompts and technical specifications for each component are provided in Appendix E. Our experimental setup is fully specified in Section 4.1, including hardware environment, time constraints, and evaluation protocol on MLE-Bench. All reported results represent mean ± SEM across three independent runs with different random seeds to ensure statistical robustness. Upon acceptance, we will publicly release the complete codebase with documentation and tutorials.

## ETHICS STATEMENT

Our work on R&D-Agent is guided by a commitment to contribute to society and human well-being by responsibly augmenting data science practice rather than replacing expert judgment. We uphold high standards of scientific excellence through rigorous evaluation on public, appropriately licensed datasets, reproducible ablations, and transparent reporting of assumptions, limitations, and negative results. To avoid harm, we do not use sensitive or personally identifiable data, we encourage domain-appropriate oversight for any deployment, and we design workflows that minimize misuse and leakage of credentials or private materials. We are honest, trustworthy, and transparent about methods, data, and outcomes; we seek fairness and take action to avoid discrimination by monitoring for bias and refraining from using protected attributes in ways that could lead to disparate impact. We respect the work required to produce new ideas and artefacts through proper citation and license compliance, and we respect privacy and honour confidentiality by safeguarding any proprietary assets shared in evaluation and by preventing unauthorized disclosure.

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

APPENDIX

## A  RELATED WORK

Recent advances in large language models (LLMs) have enabled the development of general-purpose agents capable of performing complex reasoning, planning, and decision-making across a wide range of domains (Achiam et al., 2023; Team et al., 2023; Liu et al., 2024). In the context of data-science(Chan et al., 2024; Jing et al., 2024; Majumder et al., 2024; Sahu et al., 2024; Huang et al., 2023), these agents have demonstrated the potential to markedly improve efficiency and effectiveness across diverse tasks, surpassing traditional automated methods in several benchmark evaluations. Machine learning engineering (MLE) is a rapidly growing subfield of data science, which directly delivers runnable machine learning solutions, MLE-Bench(Chan et al., 2024) is the most widely adopted for evaluating general LLM-based MLE agents, as it draws from real Kaggle competitions and incorporates human expert solutions for direct performance comparison, offering both realism and rigor.

Research into automating MLE workflows has progressed along two complementary directions. The first comprises highly encapsulated AutoML frameworks such as PyCaret (Ali, 2020) and AutoGluon (Erickson et al., 2020; Tang et al., 2024), which offer predefined modeling pipelines and automated hyperparameter optimization. The second involves LLM-driven AutoML agents that leverage the reasoning and coding abilities of LLMs to dynamically design and refine machine learning solutions. Some early methods (Jing et al., 2024) enabled agents to iteratively improve a solution within a fixed scaffold, achieving promising results in a small set of concrete scenarios, but they failed to generalize to broader MLE tasks. Therefore, a series of subsequent methods aimed at general MLE have emerged. Examples include AIDE (Jiang et al., 2025), ML-Master (Liu et al., 2025), KompeteAI (Kulibaba et al., 2025), and MLE-STAR (Nam et al., 2025), which differ in their exploration path structures, coding workflows, and reasoning pipelines. Framework-oriented efforts, such as AIRA (Toledo et al., 2025), show how flexible design space exploration can support adaptive reasoning strategies. More recently, closed-source systems like Neo (HeyNeo Team, 2025) and InternAgent (data science version) (Team et al., 2025) have achieved state-of-the-art performance, but provide little transparency about their internal designs.

These explorations show the need for a comprehensive, extensible framework for systematic design space exploration. We introduce R&D-Agent, which unifies and generalizes prior MLE agent designs by separating the strategic research phase from the tactical development phase, enabling diverse strategy integration and achieving state-of-the-art MLE-Bench results under stricter time limits.

## B  LLM USAGE STATEMENT

We used LLMs solely as general-purpose writing assistants to improve grammar, refine sentence structure, and ensure style consistency throughout the manuscript. The LLMs did not contribute to the core research ideas, framework design, experimental methodology, or interpretation of results. All research contributions, including the R&D-Agent framework design, experimental setup, and analysis, were developed entirely by the authors. The use of LLMs in our experiments (as the backend for R&D-Agent) is part of the research methodology itself and is fully documented in the experimental sections.

## C  ADDITIONAL EXPERIMENTS

### C.1  PERFORMANCE ANALYSIS ACROSS DIFFERENT BACKEND LLMS

We evaluated R&D-Agent's adaptability using three LLM configurations: GPT-4.1 only, o3 only, and hybrid o3(R)+GPT-4.1(D). Figure 4 presents the results on MLE-Bench.

The results demonstrate two key findings. First, o3 only significantly outperforms GPT-4.1 only, confirming that reasoning-enhanced models are critical for autonomous ML engineering. Second and more importantly, the hybrid configuration achieves 29.3% Any-Medal Rate, surpassing both single-model setups. This 4.0 percentage point improvement over o3 only indicates that our dual-phase architecture creates synergistic benefits beyond simple model substitution, assigning reasoning models to research and code generators to development yields a 57% relative improvement over GPT-4.1 only.

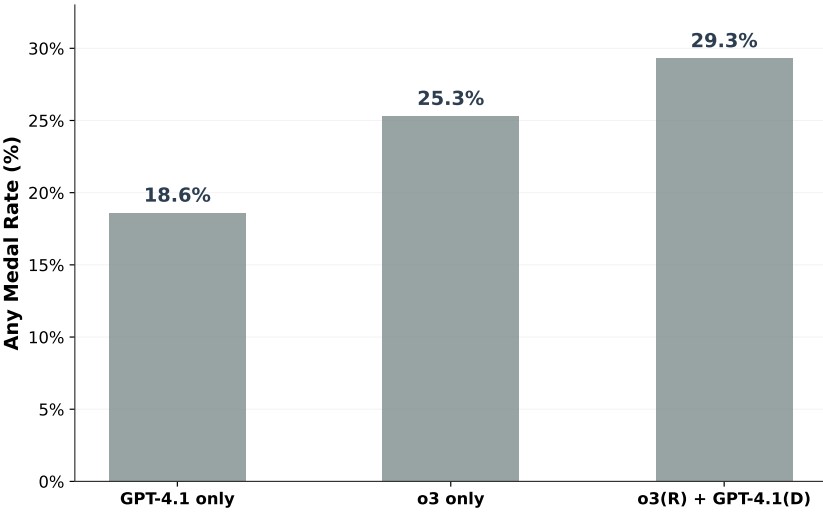

Figure 4: Performance comparison of R&D-Agent across different backend LLM configurations. The hybrid configuration achieves superior performance by leveraging specialized models for each phase.

These findings validate our framework's core design principle: the modular architecture transforms the limitation of single-model deployment into an advantage through phase-specific optimization. Each phase leverages the most suitable model capabilities, enabling both superior performance and cost-effective deployment where expensive reasoning models are selectively applied. The consistent performance hierarchy (GPT-4.1 only < o3 only < hybrid) confirms that gains stem from principled architectural design rather than model-specific tuning.

### C.2 EFFECT OF EXTERNAL KNOWLEDGE

Retrieval-Augmented Generation (RAG) has emerged as a promising approach for enhancing LLM-based agents. We investigated how incorporating external knowledge influences our agent's performance across different competition difficulty levels.

We compiled a comprehensive knowledge base from 85 Kaggle competitions (excluding those in MLE-Bench), containing high-quality notebooks and forum discussions covering diverse competition types including tabular data, computer vision, NLP, and time series forecasting. Our retrieval strategy employs embedding-based similarity matching to identify the most relevant knowledge during the research phase, where external knowledge is retrieved after problem analysis and incorporated as reference material during solution design.

Table 4: Effect of external knowledge integration on agent performance. Values represent Any-Medal rates (%) across three runs.

| Configuration | Low==Lite | Medium | High | Overall |
|---|---|---|---|---|
| R&D-Agent (baseline) | **68.2** | 21.1 | 22.2 | **35.1** |
| R&D-Agent w/ RAG | 54.6 | 21.1 | **26.7** | 32.0 |

As shown in Table 4, incorporating external knowledge surprisingly harms overall performance, with particularly severe degradation on Low==Lite tasks. This counterintuitive finding challenges the prevailing assumption that RAG universally improves LLM capabilities (Mansurova et al., 2024; Liang et al., 2025).

The only scenario where RAG proves beneficial is for high-difficulty competitions, suggesting that external knowledge primarily adds value when facing genuinely novel or specialized challenges beyond the model's training distribution. For standard ML tasks, we hypothesize that modern LLMs have already internalized common patterns sufficiently well, and external retrieval introduces

noise that disrupts their problem-solving flow. These findings suggest that RAG should be applied selectively based on task complexity rather than as a universal enhancement for MLE agents.

## C.3 COMPUTATIONAL EFFICIENCY ANALYSIS

We evaluate R&D-Agent's computational efficiency compared to existing methods. Table 5 summarizes the runtime and GPU requirements across different agents.

Table 5: Runtime and GPU Specifications of Different Agents. None indicates that the GPU information was not explicitly stated in the original paper.

| Agent | Runtime (h) | GPU |
|---|---|---|
| MLAB GPT-4o | 24 | NVIDIA A10 |
| OpenHands GPT-4o | 24 | NVIDIA A10 |
| AIDE GPT-4o | 24 | NVIDIA A10 |
| AIDE o1-preview | 24 | NVIDIA A10 |
| ML-Master Deepseek-R1 | 12 | NVIDIA A100 |
| KompeteAI Gemini-2.5-flash | 6 | NVIDIA A100 |
| MLE-STAR Gemini-2.5-pro | 24 | 8× NVIDIA V100 |
| MLE-STAR Gemini-2.0-flash | 24 | 8× NVIDIA V100 |
| AIRA Greedy | 24 | NVIDIA H200 |
| AIRA MCTS | 24 | NVIDIA H200 |
| **R&D-Agent GPT-5(ours)** | **12** | **NVIDIA V100** |

R&D-Agent achieves state-of-the-art performance using only a single NVIDIA V100 GPU within 12 hours, demonstrating superior resource efficiency compared to methods requiring multiple GPUs (e.g., MLE-STAR with 8×V100) or extended runtimes (24 hours for most baselines). The efficiency advantage that we achieve comparable or better results with 2× less time and 8× fewer GPUs makes our framework significantly more practical for real-world deployment where computational resources are constrained.

## C.4 COMPARISON WITH CLOSED-SOURCE SYSTEMS

To assess R&D-Agent's competitiveness beyond open-source baselines, we compare against recent closed-source commercial systems on MLE-Bench. As a framework for autonomous ML engineering, R&D-Agent faces the dual challenge of achieving fully autonomous operation while competing with proprietary systems that may leverage private datasets, custom infrastructure, and undisclosed optimizations. Table 6 presents the comparison results.

Table 6: Comparison with Recent Closed-Source MLE-Bench Agents. Values show mean ± SEM.

| Agent | Time (h) | Low/Lite (%) | Medium (%) | High (%) | All (%) |
|---|---|---|---|---|---|
| InternAgent (Team et al., 2025) | 12 | 62.1± 3.0 | 26.3± 2.6 | 24.4± 2.2 | **36.4± 1.2** |
| Neo (HeyNeo Team, 2025) | 36 | 48.5± 1.5 | **29.8± 2.3** | 24.4± 2.2 | 34.2± 0.9 |
| **R&D-Agent** | **12** | **68.2± 2.6** | 21.1± 1.5 | 22.2± 2.2 | 35.1± 0.4 |

Despite being fully open-source and operating completely autonomously without human intervention, R&D-Agent achieves remarkably competitive performance against closed-source systems. The framework autonomously completes the entire ML pipeline, from problem analysis through solution implementation to evaluation, approaching InternAgent's 36.4%. This near-parity performance is particularly notable given that our autonomous agent surpasses Neo (34.2%) while requiring only one-third of the runtime (12h vs. 36h), demonstrating that efficient autonomous operation need not compromise solution quality.

Our framework particularly excels on Low==Lite tasks, where R&D-Agent achieves 68.2%, outper-forming both closed-source alternatives by substantial margins. This strong performance on founda-tional tasks validates that our architectural innovationsthe dual-phase design enabling autonomous research and development, modular components for systematic exploration, and standardized evalua-tion protocolsprovide fundamental advantages for autonomous ML engineering.

## C.5 EXTENDED ANALYSIS: MLE-BENCH LITE RESULTS

While our main experiments evaluate agents on the full MLE-Bench dataset, we additionally provide detailed comparisons on the MLE-Bench Lite subset for completeness and to facilitate comparison with methods that only report Lite results (Kulibaba et al., 2025; Nam et al., 2025). Figure 5 presents comprehensive performance comparisons across all agents that have reported results on this subset.

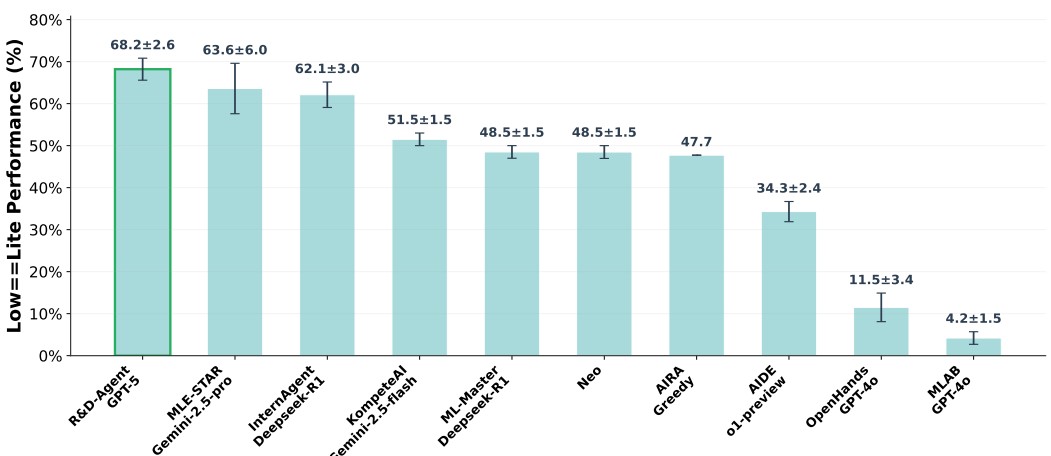

Figure 5: Agent performance on MLE-Bench (Lite). Each value represents the mean performance across all benchmark tasks, with the value after "±" indicating SEM. For the AIRA agents, the reported value is 0 because the original paper did not provide explicit results.

On the Lite subset, R&D-Agent achieves $68.2 \pm 2.6\%$, establishing the highest performance among all evaluated systems. This represents a 4.6 percentage point improvement over the previous best result of $63.6 \pm 6.0\%$ (MLE-STAR with Gemini-2.5-pro), despite using significantly fewer computational resources (as detailed in Appendix C.3). The performance progression from early agents (MLAB at $4.2 \pm 1.5\%$) to current state-of-the-art demonstrates the rapid advancement in autonomous ML engineering capabilities.

## D EXPERIMENTAL DETAILS AND SUPPLEMENTARY RESULTS

This section provides comprehensive experimental details and supplementary analysis to complement the main results, including detailed performance breakdowns and experimental design specifications.

### D.1 RAW MAIN EXPERIMENTAL RESULTS

Table 7 shows complete performance metrics for both R&D-Agent configurations across all three independent runs, providing transparency into the variability and statistical robustness of our experi-mental.

### D.2 COST ANALYSIS

One critical advantage of R&D-Agent is its exceptional cost efficiency. Table 8 presents the compu-tational costs per competition across three independent runs with GPT-5 in the main experiments, demonstrating that our framework achieves state-of-the-art performance at remarkably low costs.

At approximately $21 per competition, R&D-Agent achieves medal-winning performance at a fraction of traditional computational costs. This cost efficiency is particularly significant when compared to typical ML engineering workflows, which often require extensive hyperparameter tuning, model selection, and ensemble training that can consume hundreds or thousands of dollars in computational

Table 7: Complete performance metrics for R&D-Agent configurations across three independent runs

| Configuration | Valid Sub. (%) | Above Median (%) | Bronze (%) | Silver (%) | Gold (%) | Any Medal (%) |
|---|---|---|---|---|---|---|
| | 96.0 | 45.3 | 6.7 | 12.0 | 17.3 | 36.0 |
| R&D-Agent (GPT-5) | 96.0 | 45.3 | 4.0 | 13.4 | 17.3 | 34.7 |
| | 96.0 | 45.3 | 9.3 | 10.7 | 14.7 | 34.7 |
| | 94.7 | 45.3 | 8.0 | 5.3 | 17.3 | 30.6 |
| R&D-Agent (o3+GPT-4.1) | 93.3 | 44.0 | 5.3 | 8.0 | 16.0 | 29.3 |
| | 94.7 | 45.3 | 5.3 | 9.3 | 14.7 | 29.3 |

Table 8: Average computational cost per competition for R&D-Agent (GPT-5) across three runs (in USD)

| Run | Research Phase (per competition) | Development Phase (per competition) | Total Cost (per competition) |
|---|---|---|---|
| Run 1 | $4.68 | $11.14 | $15.82 |
| Run 2 | $7.85 | $14.37 | $22.22 |
| Run 3 | $8.43 | $15.75 | $24.18 |
| **Average** | $6.99 | $13.75 | **$20.74** |

resources. The framework's ability to autonomously complete complex ML tasks from data analysis through model development to final submission at such low costs represents a substantial reduction in the barrier to entry for competitive ML engineering. This democratization of access enables academic researchers, small organizations, and individual practitioners to engage in advanced ML development without requiring substantial computational budgets, potentially accelerating innovation across the broader ML community.

### D.3 MEMORY CONTEXT DESIGN

Existing MLE agents employ various strategies for managing information across exploration traces. MCTS-based methods achieve global information integration but often converge prematurely, while AIDE maintains independent branches with greedy exploitation followed by late-stage merging. KompeteAI combines different architectural components without explicit cross-branch communication. These approaches face a fundamental trade-off: maintaining branch independence preserves diversity but sacrifices efficiency through redundant exploration and missed knowledge transfer opportunities.

Our memory context design addresses this trade-off by enabling controlled information exchange between otherwise independent branches. While diversity emerges naturally from different initialization points and exploration paths, the lack of communication between branches leads to critical inefficiencies: (i) successful hypotheses discovered in one branch cannot inform others, making it difficult for branches to quickly reach optimal solutions; (ii) historical information from different branches cannot be fully utilized, since each branch evolves independently without access to others' past states.

To address these limitations, we introduce a probabilistic interaction mechanism inspired by statistical physics (Isihara, 2013). After generating hypotheses in each branch, we apply a probabilistic interaction kernel over all candidate hypotheses, thereby simulating the interaction process observed in physical systems.

### D.3.1 CANDIDATE HYPOTHESIS CONSTRUCTION

We construct the candidate pool from three complementary sources:

1. $h^c$: Hypotheses proposed by the current main branch, designed as solutions based on the problem in the current branch

2. $h^\star$: Globally optimal hypotheses from the highest-scoring loops in other branches (or possibly from the branch itself)

3. $h^s$: Hypotheses obtained by sampling the kernel of probabilistic interaction

The interaction kernel is formulated as:

$$U_{ij} = \alpha S_{ij} e^{-\gamma L} + \beta \tanh(\Delta_{ij}) \in [-2, 2], \quad p_{ij} = \frac{\exp(U_{ij})}{\sum_k \exp(U_{ik})}, \quad h^s \sim \text{Categorical}(p_{ij}) \quad (1)$$

where $U_{ij}$ is the interaction potential between hypothesis $h_i^c$ and all historical hypothesis $h_j$. The parameters $\alpha$ and $\beta$ are weights controlling the relative importance of the similarity $S_{ij}$ (cosine similarity between embeddings of $h_i^c$ and $h_j$) and score difference $\Delta_{ij}$. The parameter $\gamma$ is a decay factor based on the path length $L$.

The score difference $\Delta_{ij}$ is defined as:

$$\Delta_{ij} = \begin{cases} s_j^\star - s^\star, & \text{if higher score is better} \\ s^\star - s_j^\star, & \text{if lower score is better} \end{cases} \quad (2)$$

where $s^\star$ is the best score across all branches (global best), and $s_j^\star$ is the best score in the current branch. The final candidate hypotheses are $\mathcal{H}_{\text{cand}} = \{h_1^c, \ldots, h_m^c\} \cup \{h^\star\} \cup \{h_1^s, \ldots, h_n^s\}$.

This interaction potential function integrates both information from the hypothesis text and the score. The decay factor $e^{-\gamma L}$ applied to the hypothesis information reflects that the trajectory is not a Markov process, the generation of later hypotheses depends on multiple previous steps. Therefore, in the later stages of exploration, we want the weight of this component to decay rapidly, so that the score information plays a more dominant role.

### D.3.2 ADAPTIVE HYPOTHESIS SELECTION

In the second step, we use an LLM to select from these candidate hypotheses. In the LLM selection algorithm, we do not intend to strictly constrain the range of hypotheses. The provided candidate hypotheses serve only as a reference. Rather than being limited to selecting a single candidate hypothesis, the prompt suggests three possible actions: (1) **Select**: choose the best hypothesis from the candidate set; (2) **Modify**: revise an existing candidate hypothesis to improve it; (3) **Generate**: create a new hypothesis based on the candidate hypotheses. This design aims to reduce hallucinations and stabilize the outcomes across different traces.

Algorithm 2 presents the detailed selection process, which adapts its strategy based on the exploration stage and remaining time budget.

---

**Algorithm 2:** LLM-Based Hypothesis Selection

**Require:** Candidate hypotheses $\mathcal{H}_{\text{cand}} = \{h_1^c, \ldots, h_m^c\} \cup \{h^\star\} \cup \{h_1^s, \ldots, h_n^s\}$, current SOTA score $s_j^\star$, $s^\star$ global best score, time budget $T$

**Ensure:** Selected or generated hypothesis $h^o$
    // Candidate hypotheses $\mathcal{H}_{\text{cand}}$ are for reference only
    If $s_j^\star \leq s^\star$ (higher score is better), prioritize $\{h^\star\} \cup \{h_1^s, \ldots, h_n^s\}$; else prioritize $\{h_1^c, \ldots, h_m^c\}$
    **Draft Stage:** Focus on simple, quick-to-implement hypotheses
        • **Select:** Pick the most promising hypothesis from candidates
        • **Modify:** Adjust candidate (hyperparameters, loss, augmentations)
        • **Create:** Integrate advantages from multiple candidates or historical hypotheses
    **Improvement Stage:** Focus on meaningful gains without overcomplicating
        • **Select:** Pick the single most promising candidate
        • **Modify:** Refine candidate for faster iteration and improved gain
        • **Create:** Combine best parts of candidates into a new hypothesis
    **Multi-trace Merge Stage (Final):** Integrate best solutions across all traces
        • **Select:** Identify complementary solutions from different traces
        • **Modify:** Adapt solutions from other traces to current context
        • **Create:** Synthesize strengths from multiple traces into unified solution
    Return $h^o$

---

This adaptive selection mechanism, combined with the probabilistic interaction kernel, enables efficient cross-branch learning without sacrificing exploration diversity. The stage-aware strategy progresses from rapid exploration (Draft) through focused improvement to final multi-trace integration, ensuring that the system leverages the collective discoveries from all parallel explorations. The multi-trace merge stage specifically enables the synthesis of complementary solutions discovered independently, maximizing the benefit of our parallel exploration architecture.

### D.4 COMPETITION SUBSET FOR ABLATION STUDIES

Table 10 lists the complete set of 40 competitions used in our development phase ablation experiments. These competitions represent a diverse range of machine learning tasks, including classification, regression, and time-series forecasting challenges across tabular, text, image, and multimodal data types.

### D.5 COMPARATIVE BASELINE MEDAL ACHIEVEMENT STATISTICS

To provide comprehensive context for evaluating R&D-Agent's performance, we present detailed medal achievement statistics for competing baseline systems across the MLE-Bench evaluation set. This analysis examines the consistency and reliability of different agent configurations across multiple independent runs.

Table 9 summarizes the medal achievement frequency for two prominent baseline configurations: o3-4.1 and GPT-5, each evaluated across three independent runs on 75 Kaggle competitions. The results demonstrate significant variation in performance consistency across different model configurations.

Table 9: Medal achievement statistics for baseline agent configurations across 75 MLE-Bench competitions. Each configuration was evaluated across three independent runs. Values represent the number of medals achieved per competition (0-3).

| Competition | ML-Master o3+GPT4.1 | ML-Master GPT-5 | R&D-Agent GPT-5 | R&D-Agent o3+GPT4.1 |
|---|---|---|---|---|
| inaturalist-2019-fgvc6 | 0 | 0 | 3 | 3 |
| plant-pathology-2020-fgvc7 | 1 | 2 | 3 | 3 |
| h-and-m-personalized-fashion-recommendations | 2 | 0 | 0 | 0 |
| the-icml-2013-whale-challenge-right-whale-redux | 1 | 2 | 3 | 3 |
| jigsaw-toxic-comment-classification-challenge | 0 | 0 | 2 | 0 |
| detecting-insults-in-social-commentary | 3 | 1 | 3 | 3 |
| aptos2019-blindness-detection | 1 | 0 | 2 | 2 |
| iwildcam-2020-fgvc7 | 2 | 0 | 2 | 3 |
| us-patent-phrase-to-phrase-matching | 0 | 0 | 0 | 1 |
| hotel-id-2021-fgvc8 | 3 | 2 | 3 | 3 |
| freesound-audio-tagging-2019 | 0 | 0 | 0 | 0 |
| rsna-miccai-brain-tumor-radiogenomic-classification | 2 | 0 | 1 | 1 |
| text-normalization-challenge-russian-language | 0 | 1 | 3 | 2 |
| spooky-author-identification | 0 | 0 | 3 | 1 |
| tabular-playground-series-dec-2021 | 3 | 2 | 3 | 3 |
| herbarium-2022-fgvc9 | 1 | 0 | 0 | 0 |
| herbarium-2020-fgvc7 | 0 | 0 | 3 | 2 |
| plant-pathology-2021-fgvc8 | 3 | 3 | 3 | 3 |
| google-quest-challenge | 0 | 1 | 3 | 3 |
| stanford-covid-vaccine | 0 | 2 | 3 | 2 |
| random-acts-of-pizza | 0 | 3 | 3 | 1 |
| herbarium-2021-fgvc8 | 1 | 0 | 1 | 0 |
| leaf-classification | 0 | 0 | 2 | 0 |
| nomad2018-predict-transparent-conductors | 3 | 3 | 3 | 3 |
| aerial-cactus-identification | 0 | 1 | 2 | 2 |
| seti-breakthrough-listen | 1 | 0 | 1 | 1 |

**Table 9 – continued from previous page**

| Competition | ML-Master o3+GPT4.1 | ML-Master GPT-5 | R&D-Agent GPT-5 | R&D-Agent o3+GPT4.1 |
|---|---|---|---|---|
| kuzushiji-recognition | 0 | 0 | 1 | 0 |
| cassava-leaf-disease-classification | 0 | 0 | 0 | 0 |
| predict-volcanic-eruptions-ingv-oe | 2 | 2 | 3 | 3 |
| whale-categorization-playground | 0 | 0 | 0 | 2 |
| learning-agency-lab-automated-essay-scoring-2 | 0 | 0 | 1 | 0 |
| 3d-object-detection-for-autonomous-vehicles | 0 | 1 | 0 | 0 |
| histopathologic-cancer-detection | 3 | 3 | 3 | 3 |
| dogs-vs-cats-redux-kernels-edition | 3 | 2 | 3 | 3 |
| tweet-sentiment-extraction | 0 | 0 | 0 | 0 |
| mlsp-2013-birds | 0 | 0 | 1 | 0 |
| iwildcam-2019-fgvc6 | 3 | 3 | 3 | 3 |
| text-normalization-challenge-english-language | 2 | 1 | 3 | 2 |
| denoising-dirty-documents | 1 | 3 | 3 | 3 |
| google-research-identify-contrails-reduce-global-warming | 0 | 0 | 0 | 0 |
| imet-2020-fgvc7 | 0 | 0 | 0 | 0 |
| rsna-2022-cervical-spine-fracture-detection | 0 | 0 | 0 | 0 |
| vesuvius-challenge-ink-detection | 0 | 0 | 0 | 0 |
| tabular-playground-series-may-2022 | 0 | 0 | 0 | 0 |
| alaska2-image-steganalysis | 0 | 0 | 0 | 0 |
| tensorflow2-question-answering | 0 | 0 | 0 | 0 |
| dog-breed-identification | 0 | 0 | 0 | 0 |
| siim-isic-melanoma-classification | 0 | 0 | 0 | 0 |
| osic-pulmonary-fibrosis-progression | 0 | 0 | 0 | 0 |
| vinbigdata-chest-xray-abnormalities-detection | 0 | 0 | 0 | 0 |
| ranzcr-clip-catheter-line-classification | 0 | 0 | 0 | 0 |
| lmsys-chatbot-arena | 0 | 0 | 0 | 0 |
| tensorflow-speech-recognition-challenge | 0 | 0 | 0 | 0 |
| champs-scalar-coupling | 0 | 0 | 0 | 0 |
| statoil-iceberg-classifier-challenge | 0 | 0 | 0 | 0 |
| jigsaw-unintended-bias-in-toxicity-classification | 0 | 0 | 0 | 0 |
| tgs-salt-identification-challenge | 0 | 0 | 0 | 0 |
| bms-molecular-translation | 0 | 0 | 0 | 0 |
| billion-word-imputation | 0 | 0 | 0 | 0 |
| smartphone-decimeter-2022 | 0 | 0 | 0 | 0 |
| facebook-recruiting-iii-keyword-extraction | 0 | 0 | 0 | 0 |
| rsna-breast-cancer-detection | 0 | 0 | 0 | 0 |
| icecube-neutrinos-in-deep-ice | 0 | 0 | 0 | 0 |
| ventilator-pressure-prediction | 0 | 0 | 0 | 0 |
| nfl-player-contact-detection | 0 | 0 | 0 | 0 |
| siim-covid19-detection | 0 | 0 | 0 | 0 |
| hubmap-kidney-segmentation | 0 | 0 | 3 | 2 |
| hms-harmful-brain-activity-classification | 0 | 0 | 0 | 0 |
| AI4Code | 0 | 0 | 0 | 0 |
| chaii-hindi-and-tamil-question-answering | 0 | 0 | 0 | 0 |
| petfinder-pawpularity-score | 0 | 0 | 0 | 0 |
| multi-modal-gesture-recognition | 0 | 0 | 0 | 0 |
| new-york-city-taxi-fare-prediction | 0 | 0 | 0 | 0 |
| cdiscount-image-classification-challenge | 0 | 0 | 0 | 0 |
| uw-madison-gi-tract-image-segmentation | 0 | 0 | 0 | 0 |

Table 10: Complete list of 40 MLE-Bench competitions used for development phase ablation studies

| ID | Competition Name | Category | Complexity |
|---|---|---|---|
| 1 | 3d-object-detection-for-autonomous-vehicles | Image Segmentation | High |
| 2 | aerial-cactus-identification | Image Classification | Low |
| 3 | aptos2019-blindness-detection | Image Classification | Low |
| 4 | cassava-leaf-disease-classification | Image Classification | Medium |
| 5 | denoising-dirty-documents | Image to Image | Low |
| 6 | detecting-insults-in-social-commentary | Text Classification | Low |
| 7 | dogs-vs-cats-redux-kernels-edition | Image Classification | Low |
| 8 | freesound-audio-tagging-2019 | Audio Classification | Medium |
| 9 | google-quest-challenge | Training LLMs | Medium |
| 10 | h-and-m-personalized-fashion-recommendations | Tabular | Medium |
| 11 | herbarium-2020-fgvc7 | Image Classification | Medium |
| 12 | herbarium-2021-fgvc8 | Image Classification | Medium |
| 13 | herbarium-2022-fgvc9 | Image Classification | Medium |
| 14 | histopathologic-cancer-detection | Image (Other) | Low |
| 15 | hotel-id-2021-fgvc8 | Image Classification | Medium |
| 16 | hubmap-kidney-segmentation | Image Segmentation | Medium |
| 17 | inaturalist-2019-fgvc6 | Image Classification | Medium |
| 18 | iwildcam-2019-fgvc6 | Image Classification | High |
| 19 | iwildcam-2020-fgvc7 | Image Classification | Medium |
| 20 | jigsaw-toxic-comment-classification-challenge | Text Classification | Low |
| 21 | kuzushiji-recognition | Image Classification | Medium |
| 22 | leaf-classification | Image Classification | Low |
| 23 | learning-agency-lab-automated-essay-scoring-2 | Text Classification | Medium |
| 24 | mlsp-2013-birds | Audio Classification | Low |
| 25 | nomad2018-predict-transparent-conductors | Tabular | Low |
| 26 | plant-pathology-2020-fgvc7 | Image Classification | Low |
| 27 | plant-pathology-2021-fgvc8 | Image Classification | Medium |
| 28 | predict-volcanic-eruptions-ingv-oe | Signal Processing | High |
| 29 | random-acts-of-pizza | Text Classification | Low |
| 30 | rsna-miccai-brain-tumor-radiogenomic-classification | Image (Other) | High |
| 31 | seti-breakthrough-listen | Signal Processing | Medium |
| 32 | spooky-author-identification | Text Classification | Low |
| 33 | stanford-covid-vaccine | Tabular | High |
| 34 | tabular-playground-series-dec-2021 | Tabular | Low |
| 35 | text-normalization-challenge-english-language | Sequence to Sequence | Low |
| 36 | text-normalization-challenge-russian-language | Sequence to Sequence | Low |
| 37 | the-icml-2013-whale-challenge-right-whale-redux | Audio Classification | Low |
| 38 | tweet-sentiment-extraction | Text Classification | Medium |
| 39 | us-patent-phrase-to-phrase-matching | Text (Other) | Medium |
| 40 | whale-categorization-playground | Image Classification | Medium |

# E PROMPT

## E.1 PLANNING

The Planning component implements dynamic time-aware strategy selection, adapting experimental approaches based on remaining computational budget and current exploration state. Rather than relying on fixed schedules, it dynamically evaluates the exploration trace and remaining time to determine the optimal balance between breadth and depth of exploration.

---

**Competition Analysis Prompt — Planning Component**

You are a data science assistant that extracts structured information from unstructured text. The user will provide you a Kaggle competition description, and you need to extract specific details from it.
Please answer in json format with the following schema:
- **"Task Type":** The type of competition task, e.g., 'Classification', 'Regression', 'Time-Series Forecasting'
- **"Data Type":** The type of competition data, e.g., 'Tabular', 'Time Series', 'Text', 'Image', 'Audio'
- **"Brief Description":** A brief description of the competition
- **"Dataset Description":** The dataset structure based on processed data folder description
- **"Submission Specifications":** The submission specification & sample submission file descriptions
- **"Metric Evaluation Description":** A precise explanation of how submissions are scored
- **"Metric Name":** The name of the metric which this competition uses for scoring
- **"Metric Direction":** True or False as True means bigger metric number is better
- **"Longer time limit required":** True or False, whether the scenario requires a longer time limit

---

**Dynamic Expert Role Assignment — Planning Component**

You are a world-class data scientist and machine learning engineer with deep expertise in statistics, mathematics, and computer science. Your knowledge spans cutting-edge data analysis techniques, advanced machine learning algorithms, and their practical applications.
The task type for this competition is **{{ task_type }}**. The data type used in this competition is **{{ data_type }}**.
Briefly, the competition involves: {{ brief_description }}.
The evaluation metric of this competition is: {{ metric_description }}.
**Dynamic Time Management:**
- Your execution is limited to **{{ time_limit }}** when specified
- Recommended time budget: **{{ recommend_time_limit }}** for efficiency
- Leverage all computational resources during the allocated time

---

## E.2 EXPLORATION PATH STRUCTURING

The Exploration Path Structuring component manages parallel exploration across multiple solution traces, implementing intelligent merging strategies and diversity-aware selection mechanisms. It coordinates the systematic exploration of the solution space through structured branching and convergence protocols.

---

**Intelligent SOTA Selection — Exploration Path Structuring Component**

You are an expert Kaggle competitor. You are given a list of SOTA experiments and feedbacks for a Kaggle competition. You are tasked with reviewing the list of SOTA experiments and feedbacks, and selecting the most promising experiment to submit.
**Principles for Selection:**

---

1. **Valid Score as Primary Criterion:** The valid score in the feedbacks is the most crucial information and should be considered first. Also consider generalizability and risk of overfitting when scores are close.
2. **Generalizability:**
   - **Data Diversity:** Solutions leveraging more diverse data or input modalities should be favored
   - **Stable Information:** Solutions that are stable and converge faster should be prioritized
   - **Refined Representations:** Models with better generalized, robust features should be favored
3. **Risk of Overfitting:**
   - Be cautious of solutions with high valid scores that might overfit training data
   - Ensure consistent performance across different validation folds
   - Avoid significant performance fluctuations

**Output Format:**

```
{
  "selected_SOTA_idx": [positive integer or None],
  "explanation": "Brief explanation for selection"
}
```

---

### Multi-Trace Solution Merging — Exploration Path Structuring Component

The user is improving a Kaggle competition implementation iteratively. Your task is to merge multiple solutions to create a better version that combines the strengths of multiple solutions while discarding their weaknesses, to create a new version that is better than any of the given solutions alone.

**Input Structure:**
1. **Previous Main Solution:** The main solution you will build on to create an improved version
2. **Solutions to be merged:** Multiple trials of solutions that you will combine with the previous main solution. For each solution to be merged, you will receive:
   - **Solution Description:** The approach or method used in this solution
   - **Feedback to the Solution:** Steps or changes that led to success, or failure analysis

**Merging Strategy:** Systematically analyze the successful components from each solution and integrate them while avoiding known failure patterns from the feedback history.

---

### E.3  REASONING PIPELINE

The Reasoning Pipeline component orchestrates systematic hypothesis formulation through structured scientific reasoning, implementing multi-dimensional problem identification and rigorous evaluation protocols. It transforms observations and historical knowledge into testable hypotheses with quantitative assessment criteria.

---

### Systematic Problem Identification — Reasoning Pipeline Component

Your task is to analyze the provided information and identify a concise list of **Key Challenges** or **Core Problems** relevant to achieving success in this competition. Aim for **FEWER BUT BETTER** challenges (e.g., 2-3 critical challenges), focusing on the most impactful aspects.

**Core Analysis Dimensions:**
- **Gap Identification:** Examine what successful approaches highlight as unexploited methodological avenues
- **Domain-Implementation Coherence Check:** Identify technical violations of domain constraints

- **SOTA Alignment Analysis:** Compare current SOTA against dataset properties and identify discrepancies
- **Resource-Performance Trade-offs:** Identify computational or time constraint issues

**Problem Categorization Framework:**

1. **Data-Related Problems:** Missing preprocessing, feature engineering gaps, data quality issues
2. **Model-Related Problems:** Architecture misalignment, hyperparameter suboptimality
3. **Evaluation-Related Problems:** CV strategy issues, overfitting risks, metric misalignment
4. **Implementation-Related Problems:** Code bugs, inefficient implementations, timeout issues

---

**Scientific Hypothesis Generation — Reasoning Pipeline Component**

You are a research scientist formulating testable hypotheses. For each hypothesis, perform two main tasks: hypothesis proposal and rigorous five-dimensional evaluation.

**Hypothesis Development Guidelines:**

1. **Specificity & Decisiveness:** State exact, unambiguous changes. Avoid vague goals or alternatives.
2. **Testability & Actionability:** Describe implementable and measurable changes. Focus on single, unified conceptual improvements.
3. **Evidence-Based Reasoning:** Ground hypotheses in experimental history or domain knowledge.
4. **Implementation Feasibility:** Consider resource constraints and technical complexity.

**Five-Dimensional Evaluation Protocol:** Score each hypothesis (1-10) across:

- **Problem-Hypothesis Alignment:** How well the hypothesis addresses the identified problem
- **Expected Impact:** The estimated improvement after applying the hypothesis
- **Novelty:** Degree of innovation compared to previous attempts
- **Feasibility:** The ease of implementing the proposed hypothesis
- **Risk-Reward Balance:** The exploration-exploitation balance of the proposed hypothesis

**Component Classification:** Assign each hypothesis to: `DataLoadSpec`, `FeatureEng`, `Model`, `Ensemble`, or `Workflow`.

---

### E.4 MEMORY CONTEXT

The Memory Context component manages collaborative knowledge accumulation and retrieval, implementing structured feedback analysis and cross-experiment learning mechanisms. It maintains historical performance records and enables knowledge transfer across exploration traces.

---

**Structured Experiment Analysis — Memory Context Component**

You are an advanced assistant analyzing results in data-driven R&D. Your task is to analyze the current experiment's hypothesis, implementation (code and its changes), and results, explicitly comparing them with previous best SOTA result step by step.

**Step-by-step Analysis Process:**

1. **Verify Submission Format:** Check format compliance and validity
2. **Evaluate Alignment with Competition Requirements:** Assess consistency with evaluation protocol
3. **Analyze Experimental Results:** Compare performance with SOTA and validate hypothesis

**Key Analysis Components:**

- **SOTA Comparison:** Direct comparison with historical best performance

- **Code Change Analysis:** Analyze implementation differences via diff
- **Performance Evaluation:** Score comparison with metric-aware reasoning
- **Hypothesis Validation:** Whether experimental results support or refute the hypothesis

**Memory Integration Guidelines:**

1. **Historical Context:** Reference previous similar attempts and their outcomes
2. **Pattern Recognition:** Identify recurring issues or successful strategies
3. **Knowledge Transfer:** Extract reusable insights for future experiments
4. **Risk Assessment:** Evaluate potential pitfalls based on historical failures

---

Memory-Enhanced Code Generation — Memory Context Component

You are a grandmaster-level data scientist generating robust, debuggable code following systematic development process.

**Important Context:** You are working on sample datasets and your code will go through automated iterations. Design your code to be iteration-friendly with comprehensive print statements and clear debugging information to facilitate the automatic improvement process.

**Memory-Enhanced Guidelines:**

1. **Historical Learning:** Reference previous failed attempts and their feedback
2. **Pattern Reuse:** Apply successful patterns from similar tasks
3. **Error Prevention:** Avoid mistakes that occurred in previous experiments
4. **Performance Optimization:** Implement improvements suggested in historical feedback

**Quality Assurance Requirements:**

- **Debug Mode Integration:** Support -debug flag with data sampling and timing estimation
- **Structured Output:** Use print statements for progress tracking, avoid external logging dependencies
- **Reproducibility:** Implement proper random seed management and deterministic behavior
- **Resource Management:** Dynamic resource allocation and proportional data splitting

---

## E.5 Coding Workflow

The Coding Workflow component implements an efficient iterative debugging strategy that enables rapid prototyping through intelligent data sampling and systematic code evaluation. This component addresses the computational challenge of developing solutions for large-scale datasets by employing a debug-first approach that mirrors human development practices.

---

Iterative Debug Mode Implementation — Coding Workflow Component

You are a grandmaster-level data scientist generating robust, debuggable code following systematic development process.

**Important Context:** You are working on sample datasets and your code will go through automated iterations. Design your code to be iteration-friendly with comprehensive print statements and clear debugging information to facilitate the automatic improvement process.

**Debug Mode Protocol:** Your code will be executed in debug mode with the following command:

```
python main.py --debug
```

**Data Sampling Strategy:**

- **Training Data:** Sample 10% of the training data to quickly test code correctness
- **Epoch Reduction:** Run minimum epochs for rapid iteration loops
- **Test Data Efficiency:** Perform inference only on the first test sample, use placeholders for remaining samples
- **Class Preservation:** Maintain identical label class numbers between debug and full modes

**Timing and Estimation Requirements:** Implement precise timing mechanism to estimate full run duration:

```
start_time = time.time()
# Train your model (timing scope)
end_time = time.time()
debug_time = end_time - start_time
```

Output timing information in standardized format:

```
=== Start of Debug Information ===
debug_time: {actual_debug_time_in_seconds}
estimated_time: {estimated_full_run_time_in_seconds}
=== End of Debug Information ===
```

**Validation Strategy Safeguards:** Handle stratified sampling edge cases with robust fallback mechanisms:

```
try:
    fold_indices = StratifiedKFold(...).split(train_X, train_y)
except Exception as e:
    fold_indices = KFold(...).split(train_X, train_y)
```

---

**Systematic Code Evaluation Framework — Coding Workflow Component**

Rigorously evaluate code implementation through multi-stage assessment pipeline ensuring execution correctness, competition alignment, and submission authenticity.

**Evaluation Pipeline:**
1. **Execution Success:** Verify error-free code execution with focus on functionality over performance
2. **Competition Alignment:** Confirm strict adherence to evaluation rules and experimental setup consistency
3. **Debug Mode Compliance:** Validate proper debug mode implementation and timing estimation accuracy
4. **Submission Authenticity:** Verify genuine model-generated predictions, preventing fabricated or placeholder outputs

**Debug Mode Compliance Criteria:**
- **Data Sampling:** Exactly 10% training data sampling with maintained class distributions
- **Timing Accuracy:** Reasonable debug execution time with realistic full-run estimations
- **Early Stopping Integration:** Proper consideration of early stopping in time estimation calculations
- **Output Consistency:** Identical submission format between debug and full modes

**Submission Verification Protocol:**
- **Format Compliance:** Strict matching of column names, index format, and data types
- **Authenticity Verification:** Cross-reference code logic and stdout to ensure genuine model predictions
- **Anti-Cheating Measures:** Detect and reject constant, random, or hard-coded submission values
- **Model Checkpoint Usage:** Verify usage of best saved model for final predictions

**Quality Assurance Standards:** The evaluation framework enforces comprehensive quality checks including execution traceability, algorithmic appropriateness assessment, and technical implementation review to ensure reproducible and reliable solution development.

### E.6 EVALUATION STRATEGY

The Evaluation Strategy component implements automated data splitting and performance assessment protocols to ensure robust model validation and selection. This component addresses the critical challenge of overfitting to validation sets by creating consistent holdout datasets and implementing standardized evaluation protocols across experiments.

> **Automated Data Sampling — Evaluation Strategy Component**
>
> Generate a single, self-contained Python script that strictly follows the user's instructions. Requirements:
> - The script MUST be runnable via `python <file>.py` without extra arguments unless specified
> - Prefer standard libraries; it's OK to use numpy/pandas/scikit-learn if helpful
> - Use robust error handling and clear messages
> - Use relative paths only and create missing directories when needed
> - Keep the script concise and well-commented
>
> **Data Splitting Protocol:** Write a separate script based on this code to sample 90% of the data (while maintaining the class proportions as much as possible) as the new train set, and 10% as the new test set. Save the new train and test in the specified folder.
> Save test label with id to `label.csv`, which is to be used for grading. Load source data from path `./source` directory.
> Please make sure the new test set has the same columns as the original test set. Please make sure all files used in the original code and exists in source folder are also available in the specified folder.

> **Standardized Performance Grading — Evaluation Strategy Component**
>
> Write a Python script named `grade.py` to evaluate `submission.csv` produced by a model. **Evaluation Protocol:**
> - **Input files:** `label.csv` and `submission.csv` (relative to current working directory)
> - **Output format:** `"score": float, "metric": str`
> - **Metric consistency:** Use the same evaluation metric as specified in the reference code
> - **Error handling:** Implement robust parsing and validation of submission format
> - **Data splitting strategy:** Apply stratified sampling when creating train/validation splits to preserve class distribution. When certain classes have insufficient samples for proper stratification, prioritize ensuring all classes are represented in the training set, then adjust validation split accordingly
>
> The grading script must extract the competition-specific metric from the reference implementation and apply it consistently to the holdout test set, ensuring alignment between validation methodology and final evaluation criteria. The evaluation should maintain class balance awareness throughout the assessment process.

**Validation Selector Implementation:** The core evaluation strategy is implemented through the ValidationSelector class, which performs multi-candidate re-validation using consistent holdout datasets. This meta-selector operates in four stages:

1. **Candidate Collection:** Gathers top-performing experiments from multiple exploration branches using BestValidSelector with configurable candidate limits

2. **Synthetic Dataset Generation:** Creates stratified 90-10 train-test splits while preserving class proportions and data distribution characteristics

3. **Parallel Re-evaluation:** Executes all candidate models on the consistent holdout dataset using isolated execution environments

4. **Performance Ranking:** Applies standardized grading protocols to rank candidates based on holdout performance, accounting for metric direction (higher-is-better vs. lower-is-better)

This evaluation framework mitigates validation set overfitting by introducing a consistent, previously unseen test set for final model selection, while maintaining computational efficiency through parallel execution and robust error handling across the candidate pool.

# F  CASE STUDY: COMPARATIVE ANALYSIS ON JIGSAW TOXIC COMMENT CLASSIFICATION

To provide concrete evidence of R&D-Agent's capabilities, we present a detailed comparison with ML-Master on the Jigsaw Toxic Comment Classification Challenge, a multi-label text classification task with severe class imbalance. Both systems operated under identical conditions: 12-hour runtime, GPT-5 base model, and single V100 GPU. R&D-Agent achieved a bronze medal while ML-Master performed slightly above median.

Table 11 summarizes the key technical differences between the two solutions, revealing substantial distinctions in algorithmic sophistication and implementation quality.

Table 11: Technical comparison of R&D-Agent and ML-Master solutions on Jigsaw Toxic Comment Classification Challenge

| Aspect | R&D-Agent | ML-Master |
|---|---|---|
| **Performance** | Bronze Medal | Above Median |
| **Loss Function** | AsymmetricLossMultiLabel (custom) | BCEWithLogitsLoss (standard) |
| **Batch Strategy** | Adaptive (OOM fallback) | Fixed batch size |
| **Architecture** | Modified RoBERTa w/ dropout | Unmodified pretrained model |
| **Text Truncation** | Head-tail preservation | Standard truncation |
| **Debug Mode** | Integrated fast iteration | Not implemented |
| **Token Analysis** | Per-label statistics | Basic preprocessing |

Our implementation demonstrates higher code quality and algorithmic sophistication. Specifically, we design a custom `AsymmetricLossMultiLabel` loss function, which more accurately optimizes model performance in multi-label tasks with imbalanced positive and negative samples, whereas ML-Master employs the standard `BCEWithLogitsLoss`. Furthermore, we utilize the `attempt_training_with_oom_fallback` strategy, which adaptively determines the optimal batch size, thereby improving computational efficiency.

In terms of model architecture, our approach customizes `RobertaMultiLabel` by incorporating dropout and adding a flexible, trainable fully connected layer (`self.classifier(x)`), whereas ML-Master directly uses the standard `AutoModel.from_pretrained(model_name, config=self.config)` without modification. Additionally, we integrate a debug module that facilitates rapid iteration and significantly reduces development time.

Our code also implements a **Head+Tail deterministic truncation** strategy, which preserves both the beginning and end of texts, avoiding information loss, particularly for long sequences. It outputs the head/tail fraction and truncation ratio, facilitating analysis of truncation effects on model performance.

## F.1  CODE OF R&D-AGENT

**R&D-Agent: Core code (Simplified)**

```python
# This code is from R&D-agent:

# ========================
# Utility functions
# ========================
def set_global_seed(seed=42):
    random.seed(seed)
    np.random.seed(seed)
    torch.manual_seed(seed)
    torch.cuda.manual_seed_all(seed)
    torch.backends.cudnn.deterministic = True
    torch.backends.cudnn.benchmark = False

def read_csv_safely(path):
    ...
    return df

def sanitize_train_df(train_df, label_cols):
```

```python
        ...
        return train_df

def sanitize_test_df(test_df):
        ...
        return test_df

# ==========================
# Dataset & Collate
# ==========================
class TextDataset(Dataset):
    def __init__(self, texts, labels=None):
        self.texts = texts
        self.labels = labels
    def __len__(self):
        return len(self.texts)
    def __getitem__(self, idx):
        if self.labels is not None:
            return self.texts[idx], self.labels[idx]
        return self.texts[idx]

    """
    Collate function implementing deterministic head+tail truncation:
    - Reserve special tokens (tokenizer.num_special_tokens_to_add(pair=False)).
    - If content length > budget, take 75% head and 25% tail.
    - Wrap with special tokens and pad to batch max length (<= max_length).
    """
    special_tokens_count = tokenizer.num_special_tokens_to_add(pair=False)
    content_budget = max_length - special_tokens_count
    head_ratio = 0.75

    def collate(batch):
        if with_labels:
            texts, labels = zip(*batch)
        else:
            texts = batch
            labels = None

        # Minimal normalization
        texts = [str(x).strip() if x is not None else "" for x in texts]

        seqs = []
        masks = []
        for t in texts:
            content_ids = tokenizer.encode(t, add_special_tokens=False)
            if len(content_ids) <= content_budget:
                kept_ids = content_ids
            else:
                head_count = int(math.floor(head_ratio * content_budget))
                tail_count = int(content_budget - head_count)
                if tail_count <= 0:
                    kept_ids = content_ids[:content_budget]
                else:
                    kept_ids = content_ids[:head_count] + content_ids[-tail_count:]
            final_ids = tokenizer.build_inputs_with_special_tokens(kept_ids)
            seqs.append(final_ids)
            masks.append([1] * len(final_ids))

        # Pad to batch max length
        padded = tokenizer.pad(
            {"input_ids": seqs, "attention_mask": masks},
            padding=True,
            return_tensors="pt"
        )

        input_ids = padded["input_ids"]
        attention_mask = padded["attention_mask"]

        if with_labels:
            labels_tensor = torch.tensor(np.array(labels), dtype=torch.float32)
            return input_ids, attention_mask, labels_tensor
        else:
            return input_ids, attention_mask

# ==========================
# Model
# ==========================
class RobertaMultiLabel(nn.Module):
    def __init__(self, model_name, num_labels=6):
```

```python
        super().__init__()
        self.config = AutoConfig.from_pretrained(model_name)
        self.backbone = AutoModel.from_pretrained(model_name, config=self.config)
        hidden_size = getattr(self.config, "hidden_size", 768)
        dropout_prob = getattr(self.config, "classifier_dropout", 0.1)
        self.dropout = nn.Dropout(dropout_prob)
        self.classifier = nn.Linear(hidden_size, num_labels)
    def forward(self, input_ids, attention_mask):
        last_hidden = self.backbone(input_ids, attention_mask).last_hidden_state
        cls_repr = last_hidden[:, 0, :]
        x = self.dropout(cls_repr)
        logits = self.classifier(x)
        return logits

# =========================
# Loss
# =========================
class AsymmetricLossMultiLabel(nn.Module):
    def __init__(self, gamma_pos=1.0, gamma_neg=4.0, clip=0.05, eps=1e-8, reduction="mean"):
        super().__init__()
        self.gamma_pos = float(gamma_pos)
        self.gamma_neg = float(gamma_neg)
        self.clip = float(clip) if clip is not None else 0.0
        self.eps = float(eps)
        self.reduction = reduction

    def forward(self, logits: torch.Tensor, targets: torch.Tensor) -> torch.Tensor:
        # logits: [B, C], targets: [B, C] in {0,1}
        # Probabilities
        x_sigmoid = torch.sigmoid(logits)
        xs_pos = x_sigmoid
        xs_neg = 1.0 - x_sigmoid

        # Asymmetric clipping for negatives
        if self.clip > 0:
            xs_neg = torch.clamp(xs_neg + self.clip, max=1.0)

        # Log-likelihoods with numeric stability
        log_pos = torch.log(torch.clamp(xs_pos, min=self.eps))
        log_neg = torch.log(torch.clamp(xs_neg, min=self.eps))

        # Basic loss
        loss = targets * log_pos + (1.0 - targets) * log_neg

        # Asymmetric focusing
        if self.gamma_pos > 0 or self.gamma_neg > 0:
            pt = targets * xs_pos + (1.0 - targets) * xs_neg  # pt for each example/label
            one_sided_gamma = self.gamma_pos * targets + self.gamma_neg * (1.0 - targets)
            modulating = torch.pow(1.0 - pt, one_sided_gamma)
            loss = loss * modulating

        # Final reduction
        loss = -loss
        if self.reduction == "mean":
            return loss.mean()
        elif self.reduction == "sum":
            return loss.sum()
        else:
            return loss

# =========================
# Training & Evaluation
# =========================
def evaluate(model, val_loader, device, criterion, use_amp):
    ...
    return mean_auc, per_label_auc, val_loss, probs

def train_one_run(train_loader, val_loader, device, model_name, lr=2e-5, max_epochs=2, ...):
    model = RobertaMultiLabel(model_name).to(device)
    optimizer = torch.optim.AdamW(model.parameters(), lr=lr)
    scheduler = get_linear_schedule_with_warmup(optimizer, ...)
    criterion = AsymmetricLossMultiLabel()
    scaler = torch.cuda.amp.GradScaler(enabled=(device.type=="cuda"))
    best_auc = -float("inf")
    for epoch in range(max_epochs):
        model.train()
        for input_ids, attention_mask, labels in train_loader:
            ...
            scaler.scale(loss).backward()
```

```
                scaler.step(optimizer)
                scaler.update()
                scheduler.step()
          mean_auc, _, val_loss, _ = evaluate(model, val_loader, device, criterion,
          use_amp=True)
          if mean_auc > best_auc:
                best_auc = mean_auc
                best_state_dict = {k:v.cpu() for k,v in model.state_dict().items()}
    return best_state_dict, best_auc, _

def attempt_training_with_oom_fallback(train_dataset, val_dataset, tokenizer, device,
initial_batch_size=32, max_epochs=2):
    for bs in [initial_batch_size, 24, 16]:
        try:
            train_loader = DataLoader(train_dataset, batch_size=bs, shuffle=True,
            collate_fn=make_head_tail_collate_fn(tokenizer))
            val_loader = DataLoader(val_dataset, batch_size=bs, shuffle=False,
            collate_fn=make_head_tail_collate_fn(tokenizer))
            return train_one_run(train_loader, val_loader, device, model_name="roberta-base",
            max_epochs=max_epochs), bs
        except RuntimeError as e:
            if "out of memory" in str(e).lower():
                torch.cuda.empty_cache()
                continue
            else:
                raise

def main():
...
# Debug mode: sample 10% after split
    if DEBUG:
        rng = np.random.default_rng(seed)
        train_sample_size = max(1, int(0.1 * len(train_idx)))
        val_sample_size = max(1, int(0.1 * len(val_idx)))
        train_idx = rng.choice(train_idx, size=train_sample_size, replace=False)
        val_idx = rng.choice(val_idx, size=val_sample_size, replace=False)
        print(f"DEBUG mode active: using {len(train_idx)} train samples and {len(val_idx)}
        val samples (10% of split).")
...
```

## F.2 CODE OF ML-MASTER

**ML-master: Core code (Simplified)**

```
# This code is from ML master:
import torch
from torch.utils.data import Dataset, DataLoader
from transformers import AutoTokenizer, AutoModelForSequenceClassification,
get_linear_schedule_with_warmup
from torch.cuda.amp import autocast, GradScaler
import numpy as np

# === Preprocessing & data loading ===
# train_df, test_df = ...
# tokenizer = AutoTokenizer.from_pretrained("distilbert-base-uncased")
# encodings = tokenizer(...)

class ToxicDataset(Dataset):
    """Custom dataset for multi-label classification."""
    def __init__(self, encodings, labels=None):
        self.input_ids = encodings["input_ids"]
        self.attention_mask = encodings["attention_mask"]
        self.labels = labels

    def __len__(self): return len(self.input_ids)
    def __getitem__(self, idx):
        item = {
            "input_ids": torch.tensor(self.input_ids[idx]),
            "attention_mask": torch.tensor(self.attention_mask[idx]),
        }
        if self.labels is not None:
            item["labels"] = torch.tensor(self.labels[idx])
        return item

# === DataLoader setup ===
train_loader = DataLoader(ToxicDataset(...), batch_size=32, shuffle=True)
val_loader   = DataLoader(ToxicDataset(...), batch_size=32)
```

```python
# === Model, optimizer, and scheduler ===
device = torch.device("cuda" if torch.cuda.is_available() else "cpu")
model = AutoModelForSequenceClassification.from_pretrained(
    "distilbert-base-uncased", num_labels=6
).to(device)

optimizer = torch.optim.AdamW(model.parameters(), lr=2e-5)
scheduler = get_linear_schedule_with_warmup(optimizer, num_warmup_steps=...,
num_training_steps=...)
criterion = torch.nn.BCEWithLogitsLoss(pos_weight=...)
scaler = GradScaler()

# === Evaluation function ===
def evaluate(model, loader):
    """Compute ROC-AUC on validation set."""
    model.eval()
    all_probs, all_true = [], []
    with torch.no_grad():
        for batch in loader:
            input_ids = batch["input_ids"].to(device)
            mask = batch["attention_mask"].to(device)
            outputs = model(input_ids=input_ids, attention_mask=mask)
            probs = torch.sigmoid(outputs.logits).cpu().numpy()
            all_probs.append(probs)
            if "labels" in batch: all_true.append(batch["labels"].cpu().numpy())
    # Compute mean AUC over all labels
    ...
    return mean_auc

# === Training loop ===
best_auc, best_state = -np.inf, None
for epoch in range(2):
    model.train()
    for batch in train_loader:
        input_ids = batch["input_ids"].to(device)
        mask = batch["attention_mask"].to(device)
        labels = batch["labels"].to(device)

        optimizer.zero_grad(set_to_none=True)
        with autocast():
            outputs = model(input_ids=input_ids, attention_mask=mask)
            loss = criterion(outputs.logits, labels)

        scaler.scale(loss).backward()
        scaler.step(optimizer)
        scaler.update()
        scheduler.step()

    # Validation step
    val_auc = evaluate(model, val_loader)
    if val_auc > best_auc:
        best_auc, best_state = val_auc, {k: v.cpu().clone() for k, v in
        model.state_dict().items()}

# === Final evaluation & prediction ===
model.load_state_dict(best_state)
final_auc = evaluate(model, val_loader)
# test_preds = model(...)
# sub.to_csv("submission.csv")
```

