# OpenReview forum: "R&D-Agent: An LLM-Agent Framework Towards Autonomous Data Science"
_ICLR.cc/2026/Conference — ICLR 2026 Conference Withdrawn Submission_

### Official Review · Reviewer_kPLw · 2025-10-19

**Soundness:** 1
**Presentation:** 4
**Contribution:** 1
**Rating:** 2
**Confidence:** 4

**Summary:**

This paper aims to improve the capabilities of autonomous data science agents. This paper proposes R&D Agent, which consists of six main steps: (1) Planning (tricks on time savings); (2) Exploration path structuring (similar to AIDE); (3) Scientific reasoning pipeline (enhanced planning); (4) Memory context (share knowledge across branches); (5) Coding workflow (debugging based on a small subset and then coding, another trick time savings); (6) Evaluation (consistent data splits for consistent performance comparison). Then, this paper conducts experiments on MLE-Bench to demonstrate the effectiveness of the proposed framework.

**Strengths:**

- A large number of experiments on the whole set of MLE-Bench are conducted.
- The writing is of good quality.
- Ablation studies on a subset of MLE-Bench are conducted to provide the effectiveness of the proposed framework.

**Weaknesses:**

- The novelty of this paper is limited. All the contributions are more like engineering efforts and small tricks. There is no fundamental technical contribution in this paper. Also, all the techniques are already investigated by the community. I believe this paper would not bring novel technical insights for the community.

- From the perspective of empirical findings, although this paper performs large-scale experiments on MLE-Bench, the experiment design is not convincing enough for ensuring fair comparison. Here are detailed comments:

1. First of all, the comparison in Table 1 **must** build on top of the same settings and the same foundation models; otherwise, it is not correct to claim SOTA. For example, in Table 1, GLAB, OpenHands and AIDE are developed based on GPT-4o/o1-preview, while R&D-Agent is based on more advanced LLM models GPT-5. This is an unfair comparison. We cannot figure out whether the performance improvement comes from the foundation model or from the agent framework. **Furthermore**, the comparison between MLE-Master and R&D-Agent is also unfair. The prompt of ML-Master is optimized based on DeepSeek-R1; thus, we can find that ML-Master w/ DeepSeek R1 consistently outperforms other foundation models, even more advanced GPT-5. Therefore, this paper should provide empirical results of R&D-Agent with DeepSeek-R1, a consistent setting with ML-Master, to ensure fair comparison. **Last but not least**, I believe the current SOTA data science agent is MLE-STAR. However, this paper only provides comparison with MLE-STAR in the appendix and with unfair setups once again. As shown in Figure 5, R&D-Agent is built on top of GPT-5, while MLE-STAR is built on top of Gemini-2.5-pro. This is a clearly unfair comparison. Beyond this, please make sure MLE-STAR is also evaluated on the whole set of MLE-Bench, if the authors what to claim SOTA.

2. The ablation variants in Table 3 are only run once to report the results. This is clearly unacceptable, especially considering the complicated and stochastic automated data science setting. If the computational resources are limited, the authors can only conduct the experiments on MLE-Bench-Lite. But reporting results across multiple runs is a basic requirement.

- Although this paper briefly discusses related works in Table 1 and Appendix A, I think it is still necessary to comprehensively discuss related works and fundamental differences from them in a more detailed manor.

- MLE-Bench is an offline benchmark, which still suffers from the risks of data leakage. It would be better to run R&D-Agent in an online Kaggle competition to see whether the agent can really win a medal. This would be a more interesting experimental setting, compared with overly reliance on MLE-Bench.

**Questions:**

See weakness above.

---

### Official Review · Reviewer_R5xk · 2025-10-25

**Soundness:** 3
**Presentation:** 3
**Contribution:** 2
**Rating:** 4
**Confidence:** 4

**Summary:**

This paper introduces R&D Agent, an autonomous agent framework that formalizes the machine learning engineering process by explicitly separating a research phase (planning, exploration, path structuring, memory context management, and reasoning pipelines) from a development phase (coding, workflow construction, and evaluation strategy). The framework demonstrates promising performance on the MLE-bench benchmark.

**Strengths:**

1. The core idea of separating the research and development phases is interesting and conceptually aligns with how human ML practitioners operate.
2. The experimental results on MLE-bench are promising and suggest potential for structured LLM-driven engineering pipelines.

**Weaknesses:**

Deeper analysis is needed:
1. While the paper includes baseline comparisons and ablation studies, it is unclear what limits the overall system performance. Is the bottleneck primarily in the research phase (e.g., limited exploration despite the proposed reasoning techniques), or in the development phase (e.g., LLMs struggling to reliably use external libraries or tools)? A more detailed investigation could clarify this.
2. The benchmark used in the paper, although containing 40 competitions, appears to consist of only a few broad categories (e.g., classification). It would be helpful to understand whether the agent’s strategies generalize across task types. For example, what is the medal rate when grouped by task category, and are the generated methods qualitatively similar across those categories?

Discussion and Suggestions:
1. Currently, many papers propose new frameworks or agent architectures to tackle ML or reasoning tasks, often relying on similar principles such as task decomposition or multi-phase design. In this crowded space, the key contribution should not only be introducing yet another framework but also providing insightful analysis into why and how certain design choices lead to better performance.
2. In particular, deeper interpretability and diagnostic analysis would strengthen the paper. While ablation studies help isolate the effectiveness of specific modules, they often do not explain what actually changes in the agent’s behavior or generated algorithms after ablation. For example, in this paper, it remains unclear how the removal of a certain research-phase component affects the agent’s exploration strategy or code-generation quality. Providing qualitative examples or process-level comparisons would make the paper more informative and impactful.

**Questions:**

Check Weaknesses Section

---

### Official Review · Reviewer_4ouA · 2025-10-30

**Soundness:** 3
**Presentation:** 3
**Contribution:** 2
**Rating:** 2
**Confidence:** 4

**Summary:**

R&D-Agent is an agent framework for autonomous ML engineering. The main contribution is the elaborate multi-agent architecture.

**Strengths:**

Originality
- Interesting 6 component multi-agent system. Mainly an integration of existing approaches rather than new approaches

Quality
- very comprehensive and rigorous evaluations conducted

Clarity
- Well-written paper: well exposed, well documented in the appendix and clearly positioned

Significance
- Shows it outperforms other methods on ML research tasks

**Weaknesses:**

- Limited novelty: Individual components use established techniques (MCTS, tree search, memory systems, iterative debugging). This paper is also not the first to integrate many of these into agents for ML engineering and data science. e.g. the MCTS figures even look similar to Climb-DC and and the path structure mechanism like DataInterpreter

- Many missing baselines: DataInterpreter, Climb-DC, MLCopilot, LAMBDA, AutoML-Agent --- please can you outline the differences to these works

- Only closed models used (e.g. GPT-5): Use open models to also understand what’s the source of gain the foundation model or architecture.

- No failure analysis: where and why does the framework work well and when does it fail?

- Gain ablations: given the many components, more ablations should have been done to see which components contribute the most

**Questions:**

1. Can you try other models/LLMs? One needs to decouple the framework from the capabilities bestowed by the LLM. All competitors should use the same LLM to make it a like-for-like comparison of frameworks.

2. Please can you compare to the other related work (see above)

3. Does it work beyond MLE-bench type tasks? Like other DS tasks to see where it’s effeccitve and fails. Is there a specific task structure this architecture works for or is it generalisable

---

### Official Review · Reviewer_sckK · 2025-11-01

**Soundness:** 2
**Presentation:** 2
**Contribution:** 3
**Rating:** 2
**Confidence:** 4

**Summary:**

The paper claims to implement an agent system that reaches state-of-the-art performance on MLE-Bench.

**Strengths:**

The paper shows leading “Any Medal” performance on MLE-Bench when using the most recent GPT-5 model.

**Weaknesses:**

1. Regarding “Gold” performance, it still does not perform as well as ML-Master that uses Deepseek-R1 released approximately half year earlier than GPT-5. It is recommended to test the performance of the proposed agent system using Deepseek-R1 for fair comparison.

2. The paper claims “All the existing methods can be summarized as a partial optimization from our framework’s simple baseline”. It is obviously an over-claim since many existing methods were published/released earlier than this work, and those existing methods should not be considered an adaptation or optimization from this work. Instead, polite attributions to the previous existing methods should be included in this paper.

3. Continuation to its above over-claim, the paper shows limited novelty in building its agent system. Instead, a more accurate statement seems to be: Most of this paper’s key components were invented or proposed by previous methods cited or not cited in this paper. If this statement is inaccurate, please provide more details to justify.

4. The paper focuses on the optimization on MLE-Bench. However, it may be more convincing to use multiple benchmarks for evaluating the performance of the agent system. Different data science benchmarks like Economically important tasks, DA-Code and DSBench are used by different research papers and reports including OpenAI agent technical report (https://openai.com/index/introducing-chatgpt-agent/).

**Questions:**

None.

---

### Note · Authors · 2025-11-17

**Comment:**

Thank you for the valuable time and effort the reviewers put into reviewing this paper. Your suggestions are very helpful in improving our work for the new version.

After meticulously reviewing the feedback and engaging in discussions with other authors, we realized that we need to significantly revise the paper's narrative.Our work focuses on building a framework for MLE-Agents, and we are exploring innovative solutions.
But the topic is quite hot, and a framework-level narrative would overlap with other works and obscure the contributions of innovative solutions.

We agree with the reviewer's suggestion that the paper's framing could be improved to better align with its actual contributions. However, rectifying the flaws at the framework level would require a significant reorganization of the paper's content, which may not be suitable for the rebuttal period.
We have made the difficult decision to withdraw our paper in order to restructure its content. Our plan is to resubmit it to a future conference.

**Withdrawal Confirmation:**

I have read and agree with the venue's withdrawal policy on behalf of myself and my co-authors.